# PhantomWiki: On-Demand Datasets for Reasoning and Retrieval Evaluation

**Albert Gong** [* 1]  **Kamilė Stankevičiūtė** [* 1 2]  **Chao Wan** [* 1]  **Anmol Kabra** [* 1]
**Raphael Thesmar** [1]  **Johann Lee** [1]  **Julius Klenke** [1]  **Carla P. Gomes** [1]  **Kilian Q. Weinberger** [1]

## Abstract

High-quality benchmarks are essential for evaluating reasoning and retrieval capabilities of large language models (LLMs). However, curating datasets for this purpose is not a permanent solution as they are prone to data leakage and inflated performance results. To address these challenges, we propose PhantomWiki: a pipeline to generate unique and factually consistent document corpora with diverse question-answer pairs. Unlike prior work, PhantomWiki is neither a fixed dataset, nor is it based on any existing data. Instead, a new PhantomWiki instance is generated on demand for each evaluation. We vary the question difficulty and corpus size to disentangle reasoning and retrieval capabilities respectively, and find that PhantomWiki datasets are surprisingly challenging for frontier LLMs. Thus, we contribute a scalable and data leakage-resistant framework for disentangled evaluation of reasoning, retrieval, and tool-use abilities.

## 1. Introduction

Designing agents that can perform complex reasoning while interfacing with a large-scale, dynamic corpus—like Wikipedia—is a long-standing goal in the field of natural language processing (Feldman & El-Yaniv, 2019; Min et al., 2019). Such a goal may be within reach given the impressive capabilities of recent language models, which are all trained on internet-scale data. For example, the ability of LLMs to solve math problems on GSM8K (Cobbe et al., 2021) and mathematical olympiads (AlphaProof & AlphaGeometry, 2024) could bode well for agents to answer highly quantita-

tive questions. On benchmarks like DROP (Dua et al., 2019) and MMLU (Hendrycks et al., 2020), LLMs demonstrate advanced reading comprehension and general reasoning capabilities, both necessary for intelligent agents. When augmented with retrievers (Muennighoff et al., 2022) and tools (Patil et al., 2023), language models seem to already possess a strong ability for accessing external datastores and knowledge bases.

However, it is unclear to what extent these models rely on their internal knowledge, which can easily become outdated, versus their reasoning and retrieval abilities. For example, consider the question "*What is the date of birth of Wolfgang Amadeus Mozart?*". Since this fact is contained within LLM pre-training data, asking LLMs this question cannot provide reliable insight on whether the answer was deduced, retrieved or recalled. At the same time, existing approaches that perturb Wikipedia facts (Cohen et al., 2024; Meng et al., 2022; Elazar et al., 2021; Ho et al., 2020) to construct new question-answer pairs face challenges of ensuring factual consistency across articles. For example, changing Mozart's date of birth to 2025 would also require modifying Beethoven's article to erase the fact that Beethoven might have met Mozart in 1787!

One could hope to isolate reasoning from factual knowledge using mathematical or logical reasoning benchmarks. Unfortunately, such benchmarks are not entirely reliable as indicators of reasoning performance either. On GSM8K, a dataset of grade school math problems, Mirzadeh et al. (2024) report that frontier models perform significantly worse with minor or even meaningless alterations to the test data—indicating these models are vulnerable to overfitting at best and exact memorization at worst. To ensure fair comparison, LLMs need to be evaluated in a way that does not depend on any particular dataset instance.

Following this philosophy, we develop *PhantomWiki*. At the click of a button, PhantomWiki generates a synthetic, fictional universe of characters along with a set of facts about them. We reflect these facts in a large-scale corpus of templated articles, mimicking the style of fan-wiki websites. Then we generate question-answer pairs about the universe, encapsulating the types of multi-hop questions commonly considered in the question-answering (QA) literature.

---
[*]Equal contribution [1]Department of Computer Science, Cornell University, Ithaca, New York, USA [2]Department of Computer Science and Technology, University of Cambridge, Cambridge, UK. Correspondence to: Albert Gong <agong@cs.cornell.edu>, Kamilė Stankevičiūtė <ks830@cam.ac.uk>, Chao Wan <cw862@cornell.edu>, Anmol Kabra <anmol@cs.cornell.edu>.

*Proceedings of the 42nd International Conference on Machine Learning*, Vancouver, Canada. PMLR 267, 2025. Copyright 2025 by the author(s).

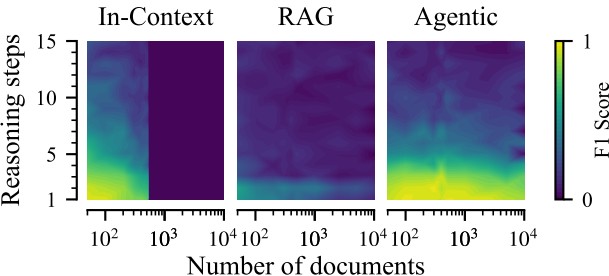

*Figure 1.* **Evaluating LLM capabilities with PhantomWiki.** We tune the reasoning and retrieval difficulty by the number of reasoning steps and documents, respectively. In this representative example of Llama-3.3-70B (other LLMs being similar), the top-right regions in all settings grow darker, indicating that the F1 score plummets as both reasoning and retrieval complexity increase. Note that in-context prompting has a sharp cut-off along the $x$-axis at Llama-3.3-70B's context length limit.

We design PhantomWiki as a data contamination-resistant tool for controlled stress-testing of LLM reasoning and retrieval capabilities. (i) On-demand, fully algorithmic generation of previously unseen universes—complete with articles and question-answer pairs—avoids the need of expensive data curation and annotation, encourages the use of fresh dataset instances, and therefore reduces the risk of data leakage. (ii) Controlling the difficulty and structure of the questions and using logic programming methods to find an exhaustive list of solutions allows for comprehensive testing of both *multi-step* and *multi-branch* reasoning, pinpointing the limits of LLM reasoning capabilities. (iii) Adjusting the universe size—and therefore the size of the reference document corpus—facilitates testing in a range of LLM settings, such as in-context learning in both short and long context windows, retrieval-augmented generation (RAG), and external tool use (as is common in agentic workflows).

Our evaluation on PhantomWiki confirms that the benchmark presents significant challenges for all state-of-the-art LLMs that we used. As a representative example, in Figure 1 we plot F1 scores for Llama-3.3-70B for in-context, RAG, and agentic settings at varying reasoning and retrieval difficulties—demonstrating the usefulness of PhantomWiki despite the simplicity of its text corpus and otherwise trivially solvable questions.

## 2. Related Work

**Question-Answering Benchmarks.** A standard technique to evaluate machine reading comprehension is through question-answering benchmarks such as SQuAD (Rajpurkar, 2016), HotpotQA (Yang et al., 2018), ComplexWebQuestions (Talmor & Berant, 2018), QAngaroo (Welbl et al., 2018), DROP (Dua et al., 2019), 2WikiMultiHopQA

(Ho et al., 2020), HybridQA (Chen et al., 2020), MuSiQue (Trivedi et al., 2022), RepLiQA (Monteiro et al., 2024) among many others. These benchmarks are typically curated by crowdsourcing questions based on excerpts from sources like Wikipedia, crowdsourcing the text corpus itself, or leveraging structured knowledge bases like Wikidata to partially automate the question-generation process (Agarwal et al., 2020; Ye et al., 2022). Such benchmark curation is limited as it is time-consuming and expensive, and the released static benchmarks are prone to eventual data contamination and memorization through internet-scraping (Trivedi et al., 2022; Monteiro et al., 2024). This has been partially addressed by dynamic frameworks such as Real-Time QA (Kasai et al., 2024) and FreshQA (Vu et al., 2023); however, those frameworks are focused on real-time factual knowledge, while PhantomWiki's focus is to evaluate knowledge-independent reasoning and retrieval.

**Retriever- and Tool-Enabled LLM Agents.** Modern LLM benchmarking places increasing importance on innovations like longer supported context lengths (Hsieh et al., 2024; Wang et al., 2024; Zhang et al., 2024; An et al., 2023; Bai et al., 2023; Li et al., 2023), the ability to reference external documents through retrieval-augmented generation (RAG) (Lewis et al., 2020; Karpukhin et al., 2020; Guu et al., 2020; Petroni et al., 2020; Saad-Falcon et al., 2023; Jin et al., 2024; Hsia et al., 2024; Mao et al., 2024; Rau et al., 2024; Shi et al., 2023; Borgeaud et al., 2022; Tang & Yang, 2024; Su et al., 2024), and the use of other external tools like search. Many of these benchmarks similarly require at least partial manual curation, do not provide a unique corpus, or, in case they are fully synthetic and customizable (Hsieh et al., 2024), are limited in their scope.

Many LLM agent benchmarks (Yao et al., 2024; Lattimer et al., 2024; Shridhar et al., 2020; Zhou et al., 2023) focus on binary-reward tasks (such as booking a flight, making a purchase), tasks that require navigation across multiple pages, or meeting user preferences. Closer to our work are tool-augmented question-answering benchmarks. For example, ToolQA (Zhuang et al., 2023) introduces a number of tasks in a range of domains and (combinations of) predefined tools (e.g. calculators and databases) to evaluate whether a model answers questions using tools or recalls a memorized answer. Compared to this work, PhantomWiki is more focused on long-lasting methods to stress-test long-context logical reasoning, uses limited tools, and resists data contamination without requiring dataset curation or manual verification of generated questions.

**Logical Reasoning Benchmarks.** Logical reasoning tasks have become central to LLM evaluation and have garnered significant attention in recent time (Zhu et al., 2023). However, many existing benchmarks do not disentangle the evaluation on logical reasoning with other abilities such as nat-

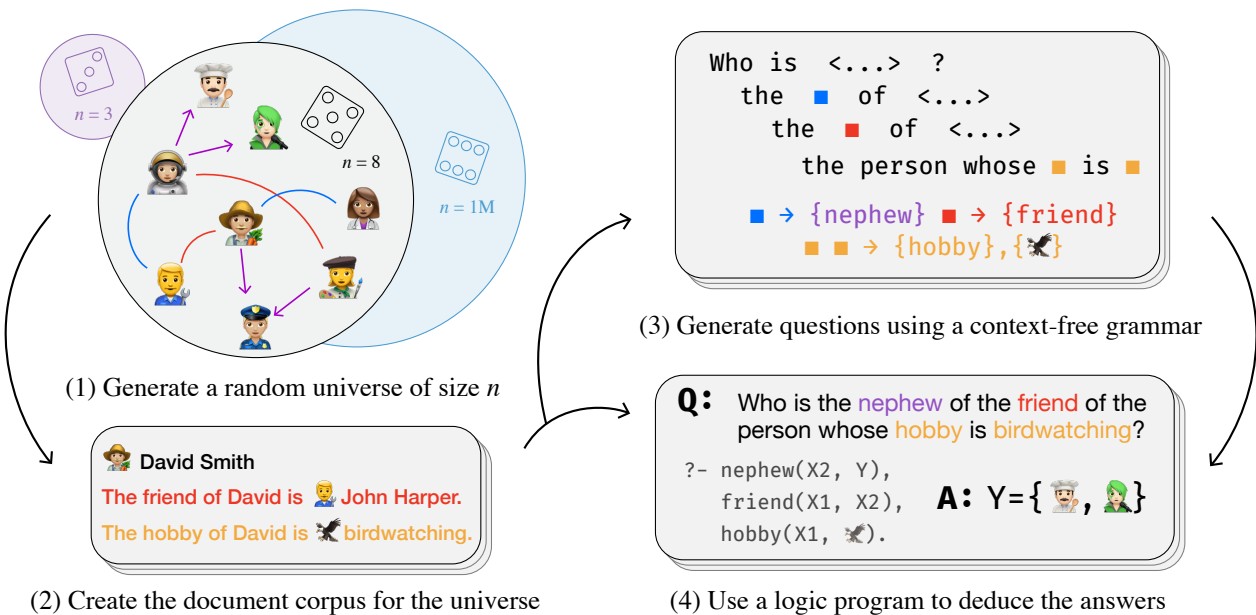

(1) Generate a random universe of size $n$

(2) Create the document corpus for the universe

(3) Generate questions using a context-free grammar

(4) Use a logic program to deduce the answers

*Figure 2.* **Overview of the PhantomWiki pipeline.**

ural language inference and commonsense reasoning (Sakaguchi et al., 2021; Zellers et al., 2019; Sprague et al., 2023). Another line of work focuses on the synthesis of datasets containing a variety of logic reasoning tasks (Tafjord et al., 2020; Saparov & He, 2022; Liu et al., 2020; Han et al., 2022; Weston et al., 2015). Closer to our work, CLUTRR (Sinha et al., 2019) crowd-source short stories about individuals related through a family graph and ask models to determine the relationship between two individuals. We extend this idea to a fully-synthetic on-demand dataset generation pipeline with more diverse question types of controllable difficulty, and enable evaluation in the modern LLM era at long-context and multi-document scale.

## 3. PhantomWiki Construction

PhantomWiki is at its core an on-demand random generator of fictional worlds. Similarly to the wiki hosting services popular in film, video games, and literature,[1] we represent these fictional worlds through Wikipedia-like biographical entries about their characters. We then test the model's retrieval skills and its understanding of the fictional world through an accompanying set of automatically generated question-answer pairs.

[1]For example, see stardewvalley.fandom.com or harrypotter.fandom.com.

### 3.1. Generating a PhantomWiki Universe

The first stage of the PhantomWiki pipeline generates a random universe of $n$ characters as well as the document corpus describing it, as illustrated in Figure 2, (1-2).

**Generating Characters.** Each character in a PhantomWiki universe is described through its *social relationships* and *personal facts* as shown in Figure 2 (1). For the social relationships, we first generate family trees, following the family tree generator of Hohenecker & Lukasiewicz (2020). We iteratively pick a person and generate their parent or child based on various constraints,[2] until the user-specified universe size of $n$ people is reached. The user can also specify other hyperparameters like the number of trees, their maximal depth, and the maximal number of offspring for each person. In addition to the family trees, we generate a friendship graph using the Erdős–Rényi model (making two people friends with some fixed probability, typically controlled by the desired average number of friendships.)

**Generating Facts.** Next, we generate personal facts for each person in the PhantomWiki universe. Names are assigned during the family generation procedure, with the first name sampled based on the character's gender and the surname based on the family tree, resulting in 15M full names in total.[3] We also add dates of birth in a way that is con-

[2]For example, the number of offspring of a person has to be smaller than some threshold, parents of the people at the maximal tree level will not be generated, etc.

[3]We use unique names in our experiments, but PhantomWiki also supports repeated names.

sistent with the existing family relations, and assign each person a job and a hobby that we uniformly sample from over 300 and 600 options respectively.

**Generating Articles.** Given all relevant facts for each person, we convert them into articles using pre-defined templates, e.g. "The job of David is a farmer. The hobby of David is birdwatching." (see Figure 2, (2)). This construction conveys the necessary information while keeping the articles short (about 160 tokens on average) and thus supports a larger effective universe size. The resulting articles are the only component of PhantomWiki available to the model during its evaluation (see Appendix E for an example of a small PhantomWiki corpus).

While it is possible to improve on the minimalistic templated articles using LLM-based rephrasing (see e.g. Shao et al. 2024), this poses the challenge of guaranteeing factual correctness without additional costs or external supervision. This has been supported by our experiments on articles rephrased using Llama-3.3-70B (see Appendix D.2)—while we did not observe any noticeable differences in performance trends, we identified factual errors (hallucinations) in the rephrased articles. Using templates therefore keeps PhantomWiki dataset cheaper (no computational or API cost), faster (no latency from LLM queries), and fully factually consistent; however we do see consistency-preserving LLM rephrasing methods as an exciting future direction.

### 3.2. Generating Question-Answer Pairs

In the second half of the PhantomWiki pipeline, we generate a set of questions with verifiable answers, as shown in Figure 2, (3-4).

**Generating Questions.** We implement automatic question generation through a context-free grammar (CFG, Hopcroft et al. 2001) of *question templates*, which we then use to sample complete questions. For example, the question template "*Who is the <relation> of <name>?*" can be used to sample the question "*Who is the friend of David?*" (see Figure 2, (3)). The main advantage of using a CFG is that it efficiently and systematically obtains *all* possible compositions of questions for some recursion depth $d$. For instance, the following subset of our context-free grammar:

$$S \rightarrow \text{Who is } R \text{ ?}$$
$$R \rightarrow \text{the } <relation> \text{ of } R'$$
$$R' \rightarrow R \mid <name>$$

can lead to questions ranging from "*Who is the friend of David?*" to "*Who is the nephew of the friend of the brother of David?*" as $d$ increases. In addition to these nested compositions, our CFG also supports questions about personal attributes (e.g. "*Who is the person whose hobby is birdwatching?*"), aggregation questions ("*How many brothers*

*does David have?*"), and combinations of all three ("*How many friends does the brother of the person whose hobby is birdwatching have?*") (See Appendix B for the full CFG.)

**Generating Answers.** To ensure that the answers to the sampled questions are verifiably correct, we represent our generated universe in Prolog, a logic programming language (Sterling & Shapiro, 1994). Each Prolog program consists of a set of facts known about the world such as `hobby("David", "birdwatching")`, and a set of rules defining how facts are related to each other, such as `nephew(X, Y) :- sibling(X, A), son(A, Y)`. The Prolog program uses these facts and rules to deduce the exhaustive set of answers to its *queries* (i.e., the CFG-generated questions). For example, a question "*Who is the nephew of the friend of the person whose hobby is birdwatching?*" corresponds to the three-statement Prolog query `?- nephew(X2, Y), friend(X1, X2), hobby(X1, "birdwatching")`, which returns all people satisfying these constraints in the PhantomWiki universe (see Figure 2 (4)).

To construct the Prolog queries automatically, we modify the CFG algorithm to generate both the question and query templates in parallel. We note, however, that the queries are separate from the final PhantomWiki corpus and question-answer pairs, and the answers returned by the Prolog program should be held out as part of the evaluation procedure.

### 3.3. PhantomWiki Complexity

The goal of PhantomWiki is to generate memorization-resistant evaluation datasets that are challenging in both reasoning and retrieval aspects. In this section, we discuss our conceptual and practical design choices that help us achieve this goal.

**Universe Space Complexity.** To ensure that our evaluation with PhantomWiki is memorization and data leakage-resistant, we first show that the space of possible universes is sufficiently large to generate enough unique instances. Observe that the number of possible friendship assignments grows at the rate of $\Theta(2^{n^2})$ (Flajolet & Sedgewick, 2009, Ex. II.5) as the number of individuals $n$ in the universe increases. Similarly, assuming each individual is assigned one fact from each category (job, hobby, etc.), the number of possible fact assignments grows at the rate $\Theta(c^n)$, where $c$ is the total number of choices across the categories. PhantomWiki thus samples a corpus from $\Theta(2^{n^2} c^n)$ possible universes, which leads to diverse datasets optimal for data leakage-resistant evaluation. We note that as future work PhantomWiki could be extended to increase this diversity, e.g. by adding a temporal dimension of events.

**Reasoning Complexity.** The CFG enables us to recursively compose templates that lead to complex reasoning ques-

tions. Observe that our CFG in Appendix B produces $\Theta(d)$ question templates as the recursion depth $d$ increases. Moreover, we can increase the difficulty of each template by increasing the number of *reasoning steps*. For example, substituting $<relation>$ with *nephew* in a template adds two reasoning steps (`nephew(X, Y) :- sibling(X, A), son(A, Y)`), since PhantomWiki articles only contain immediate family relationships like *sibling* and *son*. In contrast, substituting $<relation>$ with *second cousin* would lead to five reasoning steps. As we will show in Section 4, PhantomWiki questions are sufficiently complex to evaluate reasoning capabilities of state-of-the-art LLMs. We further note that PhantomWiki's CFG can be easily extended to support more question types like comparison and multiple-constraint questions.

**Retrieval Complexity.** To assess a model's retrieval capabilities, we increase the universe size $n$ so that the document corpus exceeds the model's context length—this makes a retriever necessary to answer questions correctly. For state-of-the-art LLMs with a context length of 128K, such as OpenAI's GPT-4o and Meta's Llama-3.3-70B, this corresponds to PhantomWiki universes of $n \gtrsim 1\text{K}$. This increases to $n \gtrsim 3\text{K}$ for Google's Gemini-1.5-Flash with context length 1M. Further scaling $n$ leads to further increase in retrieval difficulty. In Table 1, we show that PhantomWiki is well-suited for generating universes of this size on standard CPU hardware: generating questions with recursion depth $d = 10$ for size $n = 100\text{K}$—well beyond any existing LLM's context length—takes just 6 minutes on 8 Intel Cascade Lake CPU cores. Moreover, we can conveniently generate instances of $n = 1\text{M}$, which is on the scale of Wikipedia's corpus of 2 million biographical entries.[4]

*Table 1.* Runtime breakdown of generating a PhantomWiki instance for facts, articles and questions for universe sizes $n$.

| $n$ | Total Runtime | Facts | Articles | Questions |
|-----|--------------|-------|----------|-----------|
| $10^2$ | 0.97 s | 0.46 s | 0.07 s | 0.44 s |
| $10^3$ | 2.86 s | 0.90 s | 0.59 s | 1.37 s |
| $10^4$ | 20.91 s | 5.38 s | 5.87 s | 9.66 s |
| $10^5$ | 5.57 m | 0.81 m | 0.97 m | 3.79 m |
| $10^6$ | 3.86 h | 9.47 m | 11.77 m | 3.51 h |

## 4. Experimental Validation

We evaluate reasoning and retrieval capabilities of several frontier LLMs using PhantomWiki, by decomposing their performance over questions of varying difficulty and universes of varying sizes.

---

[4]https://en.wikipedia.org/wiki/Wikipedia:WikiProject_Biography, as of January 30, 2025.

### 4.1. Evaluation Setup

We generate PhantomWiki instances with $n$ ranging from 50 to 10K—a universe size for which the total length of articles exceed context lengths of all existing LLMs. For evaluation, we only provide the articles to the LLMs, not the Prolog database or the generated graphs. To ensure that our findings are not tied to any specific PhantomWiki instance, we use 3 random dataset seeds for each configuration. Creating PhantomWiki instances with different random seeds leads to entirely different combinations of names, relations, and personal facts. In each instance, we generate question templates with maximum recursion depth $d = 20$, for a total of 50 templates. We sample 10 questions for each template, yielding a total of 500 questions per PhantomWiki instance. As shown in Figures 5 and 6 (Appendix B), these questions have varying difficulty and number of answers. Accordingly, we prompt the LLMs to predict all answers as a comma-separated list and measure correctness with the answer-level F1 score.

### 4.2. Models and Prompting Techniques

We test both open- and closed-source LLMs, namely OpenAI's GPT-4o (OpenAI, 2024), Google's Gemini-1.5-Flash (Gemini Team, Google, 2024), and the instruction-tuned version of Meta's Llama-3.3-70B model (Llama Team – AI at Meta, 2024). We also evaluate DeepSeekAI's DeepSeek-R1-32B (DeepSeek AI, 2025) distilled with Qwen-2.5-32B (Qwen Team, 2024), which is an open-weights LLM trained on reasoning trace datasets. We prompt each LLM with the following techniques, broadly grouped in three ways:

**In-Context Prompting.** This technique includes the full document corpus as part of the prompt. We implement this technique with two strategies: ZEROSHOT, where the document corpus is immediately followed by the question, and Chain-of-Thought (COT) (Wei et al., 2022), where we additionally include a few examples of step-by-step reasoning leading to the correct answer. See Appendix C.1 for the prompt details.

**RAG Prompting.** This setting augments generation with a retriever as pioneered by Lewis et al. (2020). Due to its synthetic nature, the documents from PhantomWiki do not necessarily match real-world corpora, making neural retrievers a poor fit for evaluation. Instead, we use the BM25 retriever, which uses keyword matching, and search for the top-4 most relevant documents for each question. Next, we incorporate these retrieved documents into the model's prompt. Finally, we add in the same ZEROSHOT and COT prompts as In-Context Prompting. See Appendix C.2 for details about our retrieval setup.

**Agentic Prompting.** REACT (Yao et al., 2022) is a prompt-

*Table 2.* **F1 scores (in %) for various LLMs and prompting techniques.** We report mean $\pm$ standard error across 3 dataset generation seeds (except for GPT-4o with CoT and REACT due to cost constraints), and indicate the highest F1 score for each $n$ in bold. In-Context prompting is infeasible for $n = 5K$ as the corpus cannot be fully included in the context.

| Universe Size | Model | In-Context | | RAG | | Agentic |
| --- | --- | --- | --- | --- | --- | --- |
| | | ZEROSHOT | COT | ZEROSHOT-RAG | COT-RAG | REACT |
| 50 | DeepSeek-R1-32B | **42.42 ± 1.69** | **52.42 ± 2.64** | 19.93 ± 0.49 | 21.51 ± 1.31 | 5.47 ± 1.36 |
| | GPT-4o | 27.20 ± 0.76 | 50.66 | 28.05 ± 2.48 | 20.49 ± 1.07 | **38.70** |
| | Gemini-1.5-Flash | 28.49 ± 1.15 | 34.61 ± 2.41 | **28.92 ± 2.60** | 20.12 ± 1.69 | 30.92 ± 1.41 |
| | Llama-3.3-70B | 25.64 ± 0.56 | 48.37 ± 1.75 | 25.18 ± 1.91 | **27.63 ± 2.27** | 35.83 ± 1.00 |
| 500 | DeepSeek-R1-32B | **18.33 ± 2.33** | 19.65 ± 3.00 | 16.70 ± 0.85 | 17.87 ± 1.32 | 3.57 ± 0.01 |
| | GPT-4o | 16.76 ± 0.87 | **41.02** | **22.32 ± 1.99** | 16.39 ± 0.85 | **37.39** |
| | Gemini-1.5-Flash | 17.39 ± 1.45 | 25.17 ± 1.77 | 21.47 ± 1.44 | 15.09 ± 1.28 | 26.99 ± 1.84 |
| | Llama-3.3-70B | 11.59 ± 1.19 | 25.99 ± 2.09 | 19.45 ± 0.93 | **21.60 ± 2.06** | 35.56 ± 0.49 |
| 5000 | DeepSeek-R1-32B | | | 15.64 ± 0.88 | 14.81 ± 1.44 | 4.74 ± 0.04 |
| | GPT-4o | N/A (exceeds | | **18.13 ± 0.66** | 14.25 ± 1.54 | **36.85** |
| | Gemini-1.5-Flash | maximum context) | | 17.94 ± 0.93 | 12.51 ± 0.94 | 23.47 ± 1.53 |
| | Llama-3.3-70B | | | 15.07 ± 0.55 | **17.89 ± 0.45** | 30.89 ± 2.24 |

ing technique that enables LLMs to interleave reasoning steps with tool interactions, to solve complex tasks. For PhantomWiki QA task, the LLMs are provided with keyword-based tools `RetrieveArticle` and `Search` to retrieve relevant documents. See Appendix C.5 for tool details. These settings materialize the limitations of in-context prompting and necessitate the use of advanced RAG prompting and agentic prompting approaches.

In the CoT and REACT prompts, we include 10 QA exemplars and hand-written reasoning traces. We choose these exemplars from a dataset instance of size 25 that is not used for evaluation. In REACT, we limit LLMs to interact with the text corpus for up to 50 steps, which is sufficient to answer almost all questions in PhantomWiki instances.

We cap all LLM outputs to 4096 tokens and use greedy decoding (temperature = 0). For DeepSeek-R1-32B, we use temperature = 0.6 and top-$p = 0.95$, following DeepSeek AI (2025, Section 3).

### 4.3. Results

In Table 2, we report the mean F1 score across various universe sizes, LLMs, and prompting techniques. We first average F1 scores over all questions in a PhantomWiki instance, then compute the mean and standard error across the dataset generation seeds.

We first consider the small-universe setting ($n = 50$) in Table 2, which corresponds to roughly 16K tokens for the LLMs we test. In-Context prompting techniques outperform other techniques: CoT with DeepSeek-R1-32B attains the highest performance, followed by CoT with GPT-4o. Next, we consider the setting of medium universes

($n = 500$). Here the full document corpus can still be included in all LLMs' contexts, but we find that ZEROSHOT starts to perform poorly for all LLMs, and DeepSeek-R1-32B especially struggles. F1 scores of CoT for all LLMs degrade as well compared to $n = 50$, but perform comparably to REACT workflow. Finally, in the setting of large universes ($n = 5000$), none of the LLMs we evaluate can accommodate the full document corpus. As the in-context techniques are no longer sufficient, we must rely on RAG prompting and agentic prompting. RAG prompting attains poor F1 scores because the retriever fails to retrieve documents relevant for answering complex questions. On the other hand, agentic prompting technique shines in comparison to other techniques, indicating that LLMs are better suited to dynamically retrieve documents while reasoning on a question. We attribute the poor performance of DeepSeek-R1-32B with agentic prompting to its inferior tool-calling abilities compared to the other LLMs.

## 5. Evaluating Reasoning

To isolate LLM reasoning capabilities on PhantomWiki, we investigate model performance on small universes ($n = 50$) in Figure 3. Note that the context windows of all LLMs can fully include small universe document corpora. Each PhantomWiki dataset contains questions covering a wide range of difficulty. We evaluate three approaches: in-context prompting, RAG prompting, and agentic prompting. For each we plot the F1 scores as a function of question difficulty, as measured by the number of reasoning steps necessary to answer the question. As mentioned in Section 3.3, this is determined by the type of question templates and the sampled relationships. For all

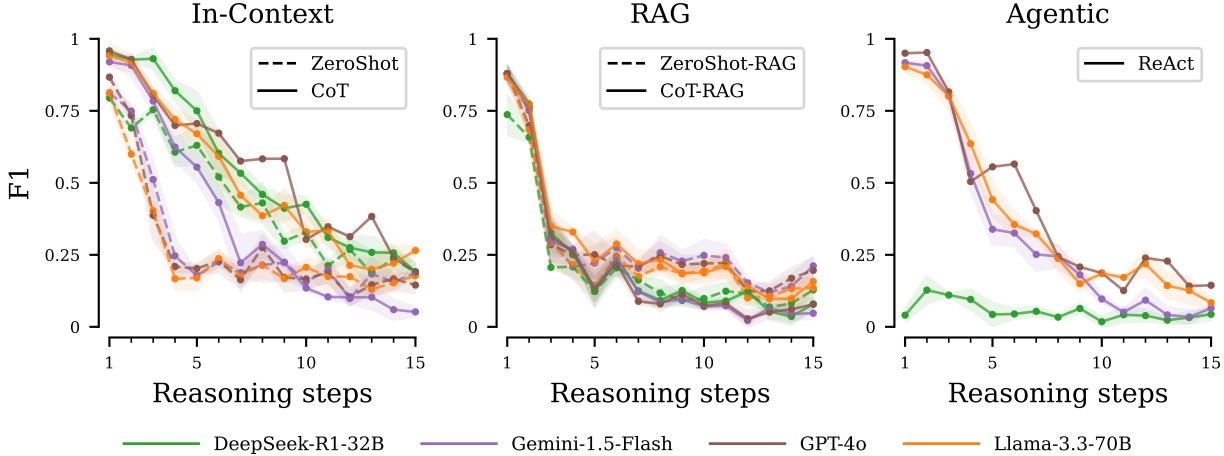

*Figure 3.* **F1 scores versus question difficulty, measured by *reasoning steps*.** We plot LLM performance on universe size $n = 50$, and report F1 scores averaged over 3 generation seeds. Increasing question difficulty in PhantomWiki reveals a clear decline across all state-of-the-art LLMs and prompting techniques, showing their struggle with reasoning.

LLMs and prompting techniques, we verify empirically that **questions with a larger number of reasoning steps are indeed more challenging to answer**.

ZEROSHOT performance declines sharply as the number of reasoning steps increases for all LLMs, except for DeepSeek-R1-32B, which deteriorates more gradually. LLMs perform better with CoT than with ZEROSHOT, but each additional reasoning step remains increasingly challenging. Another failure mode multi-branch reasoning, i.e. in keeping track of multiple reasoning traces that could lead to valid solutions. For example, a model might fail to find all possible solutions to a question "Who is the great-grandchild of David?" by forgetting to check some of the grandchildren; this compounds further as the number of possible constraints increases. This suggests that even in the absence of retrieval-specific constraints, LLMs struggle to navigate logical reasoning sequences.

RAG prompting techniques (ZEROSHOT-RAG and CoT-RAG) stunt reasoning performance across the board—F1 scores are near zero on questions with 5 or more reasoning steps as opposed to 15 steps for in-context prompting. We attribute this to a core problem with RAG prompting: retrieving documents in the initial prompt before starting to answer the question, as opposed to reasoning through the question and retrieving documents dynamically.

We find that RAG prompting techniques can only answer questions that require a single reasoning step (e.g., "Who is the friend of David?"). Thus, answering questions that require information from *multiple* reasoning steps is extremely challenging for ZEROSHOT-RAG and CoT-RAG. Consider the question, "Who is the nephew of the friend of David?," which requires retrieving David's document first,

then retrieving their friend's document to find the nephew. Since RAG prompting techniques retrieve documents *only once* by matching vector embeddings of questions and documents, they are unlikely to retrieve all necessary documents required to answer such questions. Recent multi-hop RAG prompting methods exhibit slightly better performance on PhantomWiki, but similarly struggle on questions requiring many reasoning steps, as shown in Appendix D.1.

Finally, the agentic prompting technique REACT allows LLMs to avoid the steep initial performance drop as seen in RAG prompting. On given a question, REACT prompting requires LLMs to retrieve documents dynamically in a conversation and justify why they are relevant. Concretely, before using a tool (`RetrieveArticle` or `Search`) in a conversation turn, the LLM is asked to describe how the tool will help using a "Thought" step (Yao et al., 2022), analogous to the CoT prompting approach. This approach shows promise in answering questions correctly. Even so, REACT struggles as the question difficulty increases.

Figure 3 thus decomposes LLM performance along the lines of reasoning capabilities. It reveals that all in-context prompting and agentic prompting achieve near-perfect F1 scores on low-difficulty questions. Therefore, the stratification between them in Table 2 can be attributed to varying performance on high difficulty questions. To further isolate the impact of question difficulty, in Figure 7 we plot F1 scores as a function of reasoning steps for questions with only one solution.

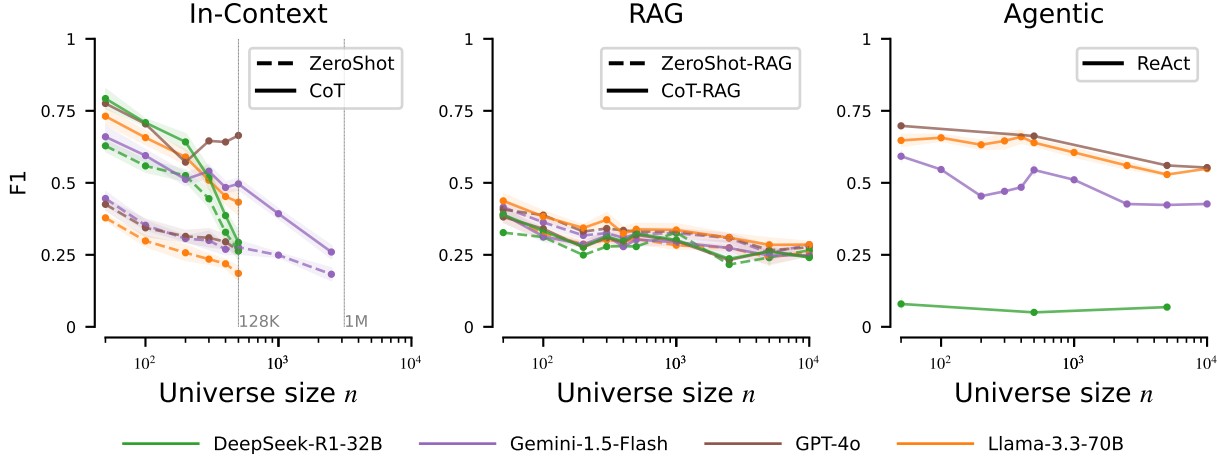

*Figure 4.* **F1 scores as a function of universe size** $n$**.** We evaluate LLM performance on questions with $\leq 10$ reasoning steps, and report F1 scores averaged over 3 dataset generation seeds. As we increase universe size in PhantomWiki, F1 scores for all LLMs and prompting techniques deteriorate—albeit at different rates.

## 6. Evaluating Retrieval

Next, to evaluate LLM retrieval capabilities, we use PhantomWiki to contrast two settings: (1) small universes where the document corpus can comfortably fit in LLM context, and (2) large universes where the full corpus exceeds context lengths. To this end, we increase the universe size up to $n = 10K$, which corresponds to document corpora well beyond the context lengths of state-of-the-art LLMs, and display the results in Figure 4.

For small universes ($n = 50$), CoT outperforms ZE-ROSHOT for all LLMs. However, F1 scores noticeably worsen as more documents are included in models' contexts, with DeepSeek-R1-32B suffering a dramatic performance drop.This analysis regime indicates that **state-of-the-art LLMs struggle at in-context retrieval** for complex question-answering tasks.

At the large universe scale, in-context prompting techniques become nonviable as the document corpus exceeds model context lengths. Therefore the use of out-of-context retrieval, such as RAG prompting and agentic prompting techniques, is necessary for obtaining the answers. Here we observe that RAG prompting techniques, whose performance is heavily determined by the quality of retrieved documents, deliver sub-par F1 scores across all universe sizes. As expected, the performance declines with increasing universe size. Interestingly, chain-of-thought does not always improve performance over zeroshot in the RAG setting. In fact, Gemini-1.5-Flash and GPT-4o with CoT-RAG underperforms their ZEROSHOT-RAG counterparts. When chain-of-thought does provide an improvement, however, the gap between CoT and zeroshot is narrower in the RAG setting than in the in-context setting. Thus, we observe

that ZEROSHOT-RAG outperforms ZEROSHOT, but CoT-RAG underperforms CoT (assuming the corpus can fit into context). Agentic prompting techniques like REACT show immense promise by avoiding a steep downward trend. This suggests that **agentic workflows can be effective in dynamically retrieving documents at scale**.

## 7. Fine-tuning on PhantomWiki

PhantomWiki generates on-demand datasets with random facts and relationships, ensuring that each dataset instance is unique and evaluation is resistant to fact memorization. Even so, it is possible that fine-tuning on PhantomWiki-generated datasets could improve performance by leveraging linguistic structure in PhantomWiki articles and questions. We fine-tune LLMs on PhantomWiki datasets, simultaneously (1) assessing the viability of PhantomWiki for training LLMs and (2) testing the robustness of PhantomWiki to memorization when evaluating LLMs.

We fine-tune Qwen2.50.5B and 3B LLMs (Qwen Team, 2024) on 10 new PhantomWiki dataset instances with two popular fine-tuning algorithms: group relative policy optimization (GRPO) (DeepSeek AI, 2024) and supervised fine-tuning (SFT), and evaluate on the three PhantomWiki datasets of size $n = 50$ used in Table 2 (refer to Appendix D.3 for details). We find that GRPO improves F1 scores over in-context prompting techniques (Table 3) and SFT does not. The performance is still far from optimal, however, and the F1 scores decline as the number of reasoning steps increases (Figure 10), imitating the phenomenon in Figure 3. This demonstrates that PhantomWiki evaluation is robust to LLM memorization, and paves the way for further research in improving LLM reasoning capabilities.

*Table 3.* **Evaluation F1 scores (in %) of LLMs fine-tuned on PhantomWiki.** As in Table 2, we report the mean $\pm$ standard error across 3 dataset generation seeds for universe size $n = 50$. ZEROSHOT and COT are prompting methods; SFT and GRPO are fine-tuning methods. Qwen2.5-0.5B COT performance is notably worse than ZEROSHOT since the small model fails to follow the desired answer template.

| Method | Qwen2.5-0.5B | Qwen2.5-3B (LoRA) |
|---|---|---|
| ZEROSHOT | $11.78 \pm 0.94$ | $16.82 \pm 2.37$ |
| COT | $2.68 \pm 0.22$ | $13.71 \pm 0.81$ |
| SFT | $11.71 \pm 1.10$ | $16.89 \pm 2.22$ |
| GRPO | $13.25 \pm 0.93$ | $31.38 \pm 0.86$ |

## 8. Conclusion and Future Work

We introduce PhantomWiki—a framework for benchmarking the reasoning and retrieval capabilities of language models. As we increase the question complexity and universe size, we observe that current state-of-the-art LLMs struggle in terms of both reasoning and retrieval. PhantomWiki is scalable and memorization-resistant, hence well-suited to evaluate future generations of language models.

Our work brings forth several research directions. One of them could be to extend PhantomWiki from its limited setting and form to support more complex types of facts and relations. The simplicity and minimalism of the text corpus could be improved using LLM-based paraphrasing methods, which, as noted before, would require innovation in preserving consistency of and preventing hallucinations in the data. In this work we focus on question-answering over text corpora; this leaves potential to extend PhantomWiki to other knowledge bases and modalities such as vision and audio, enabling analogous test suites in multimodal settings.

### Software and Data

The source code for this work can be found at github.com/kilian-group/phantom-wiki and via `pip install phantom-wiki`, and the sample Hugging-Face datasets are available at kilian-group/phantom-wiki-v1.

### Acknowledgements

KS gratefully acknowledges support from AstraZeneca. CW is supported by the National Science Foundation (NSF) OAC-2118310 and NSF-1934714 grant. This work was partially supported by funding from NewYork-Presbyterian for the NYP-Cornell Cardiovascular AI Collaboration, the National Institute of Food and Agriculture (USDA/NIFA), the Air Force Office of Scientific Research (AFOSR), and a Schmidt AI2050 Senior Fellowship, a Schmidt Sciences program. The authors thank anonymous reviewers for their helpful feedback on this work.

## Impact Statement

By leveraging context-free grammars and Prolog, PhantomWiki is able to generate large, durable and challenging datasets without using LLMs. The datasets have low computational, monetary, and environmental cost and our open-source framework is accessible to any user.

Since PhantomWiki randomly generates datasets that do not reference any existing data, the evaluation benchmark is resistant to data leakage and memorization while training. The approach of publishing a dataset generation *procedure* rather than a fixed dataset also encourages better research practices (by using fresh datasets instead of overfitting to a single instance), and enables a more accurate evaluation of model performance. Since we do not use any personal data, use of PhantomWiki does not have any privacy concerns.

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

# A. Background

## A.1. Context-Free Grammars

Context-free grammar (CFG) is a type of formal grammar where the productions rules govern how to generate text from non-terminals and terminals. A context-free grammar is defined by $G = (V, \Sigma, R, S)$ where $V$ and $\Sigma$ denotes nonterminal and terminal respectively. $R$ is a finite relation in $V \times (V \cup \Sigma)^*$ which specifies the production rules of the grammar. $S \in V$ is the start symbol. A production rule in $R$ has the form

$$\alpha \to \beta \tag{1}$$

where $\alpha \in V$, $\beta \in (V \cup \Sigma)^*$. It is conventional to list all rules with the same left-hand side on the same line and separate the right-hand side with "|" like $\alpha \to \beta_1 \,|\, \beta_2$.

# B. Question Template Generation

## B.1. Context-Free Grammar

We use the following CFG to generate question templates:

```
S -> Who is R? | What is A ? | How many RN_p does R_c have ?
R -> the RN of R_c | the person whose AN is AV
R_c -> R | N
A -> the AN of R
RN -> <relation>
RN_p -> <relation_plural>
AN -> <attribute_name>
AV -> <attribute_value>
N -> <name>
```

## B.2. CFG-generated question templates

Our CFG produces the following 50 question templates at recursion depth $d = 20$. Note how the recursive production rule `R_c -> R | N` leads to chained productions.

1. Who is the <relation>_3 of the <relation>_5 of the <relation>_7 of the < relation>_9 of the <relation>_11 of the <relation>_13 of the <relation>_15 of the <relation>_17 of the person whose <attribute_name>_19 is < attribute_value>_19?
2. Who is the <relation>_3 of the <relation>_5 of the <relation>_7 of the < relation>_9 of the <relation>_11 of the <relation>_13 of the <relation>_15 of the <relation>_17 of <name>_18?
3. Who is the <relation>_3 of the <relation>_5 of the <relation>_7 of the < relation>_9 of the <relation>_11 of the <relation>_13 of the <relation>_15 of the person whose <attribute_name>_17 is <attribute_value>_17?
4. Who is the <relation>_3 of the <relation>_5 of the <relation>_7 of the < relation>_9 of the <relation>_11 of the <relation>_13 of the <relation>_15 of <name>_16?
5. Who is the <relation>_3 of the <relation>_5 of the <relation>_7 of the < relation>_9 of the <relation>_11 of the <relation>_13 of the person whose < attribute_name>_15 is <attribute_value>_15?
6. Who is the <relation>_3 of the <relation>_5 of the <relation>_7 of the < relation>_9 of the <relation>_11 of the <relation>_13 of <name>_14?
7. Who is the <relation>_3 of the <relation>_5 of the <relation>_7 of the < relation>_9 of the <relation>_11 of the person whose <attribute_name>_13 is < attribute_value>_13?
8. Who is the <relation>_3 of the <relation>_5 of the <relation>_7 of the < relation>_9 of the <relation>_11 of <name>_12?

9. Who is the <relation>_3 of the <relation>_5 of the <relation>_7 of the < relation>_9 of the person whose <attribute_name>_11 is <attribute_value>_11?
10. Who is the <relation>_3 of the <relation>_5 of the <relation>_7 of the < relation>_9 of <name>_10?
11. Who is the <relation>_3 of the <relation>_5 of the <relation>_7 of the person whose <attribute_name>_9 is <attribute_value>_9?
12. Who is the <relation>_3 of the <relation>_5 of the <relation>_7 of <name>_8?
13. Who is the <relation>_3 of the <relation>_5 of the person whose < attribute_name>_7 is <attribute_value>_7?
14. Who is the <relation>_3 of the <relation>_5 of <name>_6?
15. Who is the <relation>_3 of the person whose <attribute_name>_5 is < attribute_value>_5?
16. Who is the <relation>_3 of <name>_4?
17. Who is the person whose <attribute_name>_3 is <attribute_value>_3?
18. What is the <attribute_name>_3 of the <relation>_4 of the <relation>_6 of the <relation>_8 of the <relation>_10 of the <relation>_12 of the <relation>_14 of the <relation>_16 of the <relation>_18 of <name>_19?
19. What is the <attribute_name>_3 of the <relation>_4 of the <relation>_6 of the <relation>_8 of the <relation>_10 of the <relation>_12 of the <relation>_14 of the <relation>_16 of the person whose <attribute_name>_18 is < attribute_value>_18?
20. What is the <attribute_name>_3 of the <relation>_4 of the <relation>_6 of the <relation>_8 of the <relation>_10 of the <relation>_12 of the <relation>_14 of the <relation>_16 of <name>_17?
21. What is the <attribute_name>_3 of the <relation>_4 of the <relation>_6 of the <relation>_8 of the <relation>_10 of the <relation>_12 of the <relation>_14 of the person whose <attribute_name>_16 is <attribute_value>_16?
22. What is the <attribute_name>_3 of the <relation>_4 of the <relation>_6 of the <relation>_8 of the <relation>_10 of the <relation>_12 of the <relation>_14 of <name>_15?
23. What is the <attribute_name>_3 of the <relation>_4 of the <relation>_6 of the <relation>_8 of the <relation>_10 of the <relation>_12 of the person whose < attribute_name>_14 is <attribute_value>_14?
24. What is the <attribute_name>_3 of the <relation>_4 of the <relation>_6 of the <relation>_8 of the <relation>_10 of the <relation>_12 of <name>_13?
25. What is the <attribute_name>_3 of the <relation>_4 of the <relation>_6 of the <relation>_8 of the <relation>_10 of the person whose <attribute_name>_12 is <attribute_value>_12?
26. What is the <attribute_name>_3 of the <relation>_4 of the <relation>_6 of the <relation>_8 of the <relation>_10 of <name>_11?
27. What is the <attribute_name>_3 of the <relation>_4 of the <relation>_6 of the <relation>_8 of the person whose <attribute_name>_10 is <attribute_value>_10 ?
28. What is the <attribute_name>_3 of the <relation>_4 of the <relation>_6 of the <relation>_8 of <name>_9?
29. What is the <attribute_name>_3 of the <relation>_4 of the <relation>_6 of the person whose <attribute_name>_8 is <attribute_value>_8?
30. What is the <attribute_name>_3 of the <relation>_4 of the <relation>_6 of < name>_7?
31. What is the <attribute_name>_3 of the <relation>_4 of the person whose < attribute_name>_6 is <attribute_value>_6?
32. What is the <attribute_name>_3 of the <relation>_4 of <name>_5?
33. What is the <attribute_name>_3 of the person whose <attribute_name>_4 is < attribute_value>_4?

34. How many <relation_plural>_2 does the <relation>_4 of the <relation>_6 of the <relation>_8 of the <relation>_10 of the <relation>_12 of the <relation>_14 of the <relation>_16 of the <relation>_18 of <name>_19 have?
35. How many <relation_plural>_2 does the <relation>_4 of the <relation>_6 of the <relation>_8 of the <relation>_10 of the <relation>_12 of the <relation>_14 of the <relation>_16 of the person whose <attribute_name>_18 is < attribute_value>_18 have?
36. How many <relation_plural>_2 does the <relation>_4 of the <relation>_6 of the <relation>_8 of the <relation>_10 of the <relation>_12 of the <relation>_14 of the <relation>_16 of <name>_17 have?
37. How many <relation_plural>_2 does the <relation>_4 of the <relation>_6 of the <relation>_8 of the <relation>_10 of the <relation>_12 of the <relation>_14 of the person whose <attribute_name>_16 is <attribute_value>_16 have?
38. How many <relation_plural>_2 does the <relation>_4 of the <relation>_6 of the <relation>_8 of the <relation>_10 of the <relation>_12 of the <relation>_14 of <name>_15 have?
39. How many <relation_plural>_2 does the <relation>_4 of the <relation>_6 of the <relation>_8 of the <relation>_10 of the <relation>_12 of the person whose < attribute_name>_14 is <attribute_value>_14 have?
40. How many <relation_plural>_2 does the <relation>_4 of the <relation>_6 of the <relation>_8 of the <relation>_10 of the <relation>_12 of <name>_13 have?
41. How many <relation_plural>_2 does the <relation>_4 of the <relation>_6 of the <relation>_8 of the <relation>_10 of the person whose <attribute_name>_12 is <attribute_value>_12 have?
42. How many <relation_plural>_2 does the <relation>_4 of the <relation>_6 of the <relation>_8 of the <relation>_10 of <name>_11 have?
43. How many <relation_plural>_2 does the <relation>_4 of the <relation>_6 of the <relation>_8 of the person whose <attribute_name>_10 is <attribute_value>_10 have?
44. How many <relation_plural>_2 does the <relation>_4 of the <relation>_6 of the <relation>_8 of <name>_9 have?
45. How many <relation_plural>_2 does the <relation>_4 of the <relation>_6 of the person whose <attribute_name>_8 is <attribute_value>_8 have?
46. How many <relation_plural>_2 does the <relation>_4 of the <relation>_6 of < name>_7 have?
47. How many <relation_plural>_2 does the <relation>_4 of the person whose < attribute_name>_6 is <attribute_value>_6 have?
48. How many <relation_plural>_2 does the <relation>_4 of <name>_5 have?
49. How many <relation_plural>_2 does the person whose <attribute_name>_4 is < attribute_value>_4 have?
50. How many <relation_plural>_2 does <name>_3 have?

## B.3. Question-Answer Characteristics

# C. Baseline Details

### C.1. ZeroShot-Simple

We use the following prompt for all models, where `evidence` is the concatenation of all documents in the PhantomWiki instance.

```
You are given the following evidence:
(BEGIN EVIDENCE)
{{evidence}}
(END EVIDENCE)
```

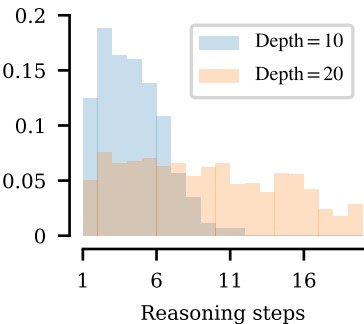

*Figure 5.* Histogram of question difficulties (measured by reasoning steps) for universe size $n = 50$ at two CFG recursion depths $d \in \{10, 20\}$. We average the frequencies across 3 dataset generation seeds.

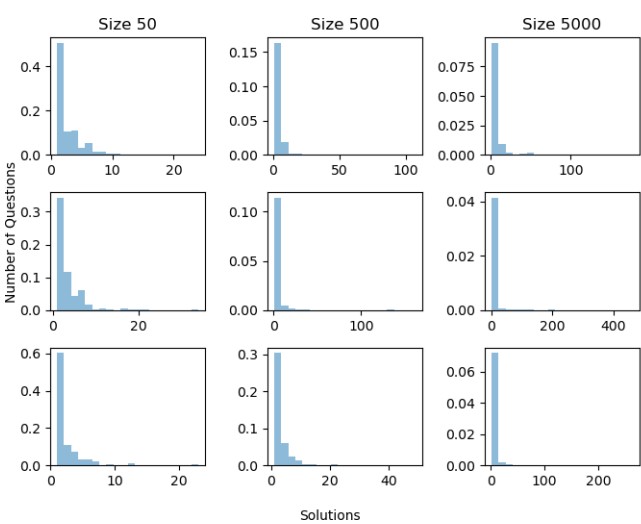

*Figure 6.* Distribution of number of answers across sizes $n \in \{50, 500, 5000\}$, seeds $\{1, 2, 3\}$, and CFG depth 20.

```
You will be provided a question. Your task is to provide an answer according to
    these instructions:
- The output must be one of the following: a name (if there is only one correct
    answer); or a list of names separated by '{constants.answer_sep}' (if there
    are multiple correct answers).
- DO NOT include any additional information in your answer.

Question: {{question}}
Answer:
```

For DeepSeek-R1-32B, we additionally parse the output to separate the model's reasoning process from its final answer using the $</think>$ tag.

## C.2. ZEROSHOT-RAG

The prompt is exactly the same as ZEROSHOT, except we replace `evidence` with 4 documents retrieved using the BM25 retriever. Upon generation, we search for similar documents for `question` according to maximum inner product search on document and question embeddings.

## C.3. CHAIN-OF-THOUGHT-SIMPLE

We use the following prompt for all models, where `evidence` is replaced with a list of all documents. We use a regular expression to parse the output.

```
You are given the following evidence:
(BEGIN EVIDENCE)
{{evidence}}
(END EVIDENCE)

You will be provided a question. Your response must end in the following sentence
    : The answer is <answer>.
Here, <answer> must be one of the following:
- a name (if there is only one correct answer); or
- a list of names separated by '{constants.answer_sep}' (if there are multiple
    correct answers).

Here are some examples:
(START OF EXAMPLES)
Example 1:
Question: Who is the brother of Dino Beltran?
Answer: Based on the evidence, the brother of Dino Beltran is Orlando Beltran.
    The answer is Orlando Beltran.

Example 2:
Question: Who is the sibling of Barabara Beltran?
Answer: Based on the evidence, the siblings of Barabara Beltran are Aida Wang,
    Vicki Hackworth. The answer is Aida Wang{constants.answer_sep}Vicki Hackworth
    .

Example 3:
Question: Who is the child of the sibling of Stacia Toombs?
Answer: First I need to find the sibling of Stacia Toombs. Based on the evidence,
     the sibling of Stacia Toombs is Shelli Beltran. Now I need to find the child
     of Shelli Beltran. Based on the evidence, the children of Shelli Beltran are
     Aida Wang, Barabara Beltran, Vicki Hackworth. The answer is Aida Wang{
    constants.answer_sep}Barabara Beltran{constants.answer_sep}Vicki Hackworth.

Example 4:
Question: Who is the uncle of William Smock?
Answer: An uncle is the brother of a parent. Based on the evidence, the parents
    of William Smock are Dominique Smock, Gene Smock. To find the uncle of
    William Smock, I need to find the brother of Dominique Smock and Gene Smock.
    Based on the evidence, Dominique Smock has no brother, and the brother of
    Gene Smock is Eli Smock. So the uncle of William Smock is Eli Smock. The
    answer is Eli Smock.

Example 5:
Question: What is the occupation of the sister of the grandmother of Virgil
    Hackworth?
Answer: A grandmother is the mother of a parent. Based on the evidence, the
    parents of Virgil Hackworth are Ricardo Hackworth, Vicki Hackworth. To find
    the grandmother of Virgil Hackworth, I need to find the mother of Ricardo
    Hackworth and Vicki Hackworth. Based on the evidence, Ricardo Hackworth has
    no mother, and the mother of Vicki Hackworth is Shelli Beltran. Now I need to
```

find the sister of Shelli Beltran. Based on the evidence, the sister of
Shelli Beltran is Stacia Toombs. Based on the evidence, the occupation of
Stacia Toombs is actuary. The answer is actuary.

Example 6:
Question: Who is the brother of the person whose occupation is associate
    professor?
Answer: I need to search for people whose occupation is associate professor.
    Based on the evidence, the person whose occupation is associate professor is
    Dino Beltran. And the brother of Dino Beltran is Orlando Beltran. The answer
    is Orlando Beltran.

Example 7:
Question: What is the date of birth of the person whose hobby is meteorology?
Answer: I need to search for people whose hobby is meteorology. Based on the
    evidence, the people whose hobby is meteorology are Alison Smock, Barabara
    Beltran. The date of birth of Alison Smock is 0929-10-28, and the date of
    birth of Barabara Beltran is 0989-06-11. The answer is 0929-10-28{constants.
    answer_sep}0989-06-11.

Example 8:
Question: Who is the cousin of the person whose occupation is broadcast engineer?
Answer: I need to search for people whose occupation is broadcast engineer. Based
     on the evidence, the person whose occupation is broadcast engineer is
    Barabara Beltran. A cousin is the child of the sibling of the parent. Based
    on the evidence, the parents of Barabara Beltran are Dino Beltran, Shelli
    Beltran. The sibling of Dino Beltran is Orlando Beltran, and the sibling of
    Shelli Beltran is Stacia Toombs. Based on the evidence, Orlando Beltran has
    no child, and the child of Stacia Toombs is Leslee Toombs. So the cousin of
    Barabara Beltran is Leslee Toombs. The answer is Leslee Toombs.

Example 9:
Question: Who is the great-granddaughter of the person whose hobby is biology?
Answer: I need to search for people whose hobby is biology. Based on the evidence
    , the person whose hobby is biology is Alvaro Smock. To find the great-
    granddaughter of Alvaro Smock, I need to find the daughter of the child of
    the child of Alvaro Smock. Based on the evidence, the children of Alvaro
    Smock are Eli Smock, Gene Smock. Eli Smock has no child, and the child of
    Gene Smock is Williams Smock. The daughters of Williams Smock are Shelli
    Beltran, Stacia Toombs. So the great-granddaughters of Alvaro Smock, whose
    hobby is biology, are Shelli Beltran, Stacia Toombs. The answer is Shelli
    Beltran{constants.answer_sep}Stacia Toombs.
(END OF EXAMPLES)

Question: {{question}}
Answer:

## C.4. CHAIN-OF-THOUGHT-RAG

The prompt is exactly the same as CoT, except we replace `evidence` with 4 documents retrieved using BM25. We use the same retriever setup as described in Appendix C.2.

## C.5. REACT

We modify the ReAct agent implementation of Shinn et al. (2024), which can be found at https://github.com/noahshinn/reflexion, to support our keyword-based search tools `RetrieveArticle` and `Search`. As seen in the following examples, the `RetrieveArticle` tool fetches the documents by title, and the `Search` tool fetches all document titles whose bodies contain the argument. We instruct the LLM to output predict the answer using a third tool `Finish`.

```
Solve a question answering task with interleaving Thought, Action, Observation
    steps.
Thought can reason about the current situation, and Action can be 3 types:
(1) RetrieveArticle[{{{{entity}}}}]. This action retrieves the article about {{{{
    entity}}}} if it exists. If the article does not exist, the action will say
    so.
(2) Search[{{{{attribute}}}}]. This action searches the database for {{{{
    attribute}}}} and retrieves all articles that contain {{{{attribute}}}}. If
    no article contains {{{{attribute}}}}, the action will say so.
(3) Finish[{{{{answer}}}}]. This action answers the question with {{{{answer}}}}.
If you cannot find the answer, output the empty answer like: Finish[].
If there are multiple answers A,B,C, answer with a list like: Finish[A{constants.
    answer_sep}B{constants.answer_sep}C].

You may take as many steps as necessary.
Here are some examples:
(START OF EXAMPLES)
Example 1:
Question: Who is the brother of Dino Beltran?
Thought 1: I need to retrieve article about Dino Beltran and find who his brother
    is.
Action 1: RetrieveArticle[Dino Beltran].
Observation 1: # Dino Beltran ## Family The parents of Dino Beltran are Brian
    Beltran, Daisy Beltran.  Dino Beltran's sibling is Orlando Beltran.  The
    brother of Dino Beltran is Orlando Beltran.  The mother of Dino Beltran is
    Daisy Beltran.  The father of Dino Beltran is Brian Beltran.  The children of
     Dino Beltran are Aida Wang, Barabara Beltran, Vicki Hackworth.  The
    daughters of Dino Beltran are Aida Wang, Barabara Beltran, Vicki Hackworth.
    The wife of Dino Beltran is Shelli Beltran.  ## Friends The friend of Dino
    Beltran is Alvaro Smock.  ## Attributes The date of birth of Dino Beltran is
    0958-08-09.  The occupation of Dino Beltran is associate professor.  The
    hobby of Dino Beltran is shogi.
Thought 2: The brother of Dino Beltran is Orlando Beltran, so the answer is
    Orlando Beltran.
Action 2: Finish[Orlando Beltran].

Example 2:
Question: Who is the mother of Ivana Smith?
Thought 1: I need to retrieve article about Ivana Smith and find who her mother
    is.
Action 1: RetrieveArticle[Ivana Smith].
Observation 1: No article exists for the requested entity. Please try retrieving
    article for another entity.
Thought 2: Article about Ivana Smith does not exist, so I cannot find the mother
    of Ivana Smith. I will output empty answer.
Action 2: Finish[].

Example 3:
```

Question: Who is the sibling of Barabara Beltran?
Thought 1: I need to retrieve article about Barabara Beltran and find who her sibling is.
Action 1: RetrieveArticle[Barabara Beltran].
Observation 1: # Barabara Beltran ## Family The parents of Barabara Beltran are Dino Beltran, Shelli Beltran. Barabara Beltran's siblings are Aida Wang, Vicki Hackworth. The sisters of Barabara Beltran are Aida Wang, Vicki Hackworth. The mother of Barabara Beltran is Shelli Beltran. The father of Barabara Beltran is Dino Beltran. ## Friends ## Attributes The date of birth of Barabara Beltran is 0989-06-11. The occupation of Barabara Beltran is broadcast engineer. The hobby of Barabara Beltran is meteorology.
Thought 2: The siblings of Barabara Beltran are Aida Wang, Vicki Hackworth. So the answer is Aida Wang, Vicki Hackworth.
Action 2: Finish[Aida Wang{constants.answer_sep}Vicki Hackworth].

Example 4:
Question: Who is the child of the sibling of Stacia Toombs?
Thought 1: First, I need to retrieve article about Stacia Toombs and find who her sibling is.
Action 1: RetrieveArticle[Stacia Toombs].
Observation 1: # Stacia Toombs ## Family The parents of Stacia Toombs are Alison Smock, Williams Smock. Stacia Toombs's sibling is Shelli Beltran. The sister of Stacia Toombs is Shelli Beltran. The mother of Stacia Toombs is Alison Smock. The father of Stacia Toombs is Williams Smock. The child of Stacia Toombs is Leslee Toombs. The daughter of Stacia Toombs is Leslee Toombs. The husband of Stacia Toombs is Wilbert Toombs. ## Friends The friends of Stacia Toombs are Brian Beltran, Isiah Lutz, Leeann Hackworth, Lesley Lutz, Ryan Wang. ## Attributes The date of birth of Stacia Toombs is 0959-03-22. The occupation of Stacia Toombs is actuary. The hobby of Stacia Toombs is finance.
Thought 2: The sibling of Stacia Toombs is Shelli Beltran. Now I need to retrieve article about Shelli Beltran and find who her child is.
Action 2: RetrieveArticle[Shelli Beltran].
Observation 2: # Shelli Beltran ## Family The parents of Shelli Beltran are Alison Smock, Williams Smock. Shelli Beltran's sibling is Stacia Toombs. The sister of Shelli Beltran is Stacia Toombs. The mother of Shelli Beltran is Alison Smock. The father of Shelli Beltran is Williams Smock. The children of Shelli Beltran are Aida Wang, Barabara Beltran, Vicki Hackworth. The daughters of Shelli Beltran are Aida Wang, Barabara Beltran, Vicki Hackworth. The husband of Shelli Beltran is Dino Beltran. ## Friends The friends of Shelli Beltran are Brian Beltran, Eli Smock, Isiah Lutz, Leslee Toombs, Lesley Lutz, Ryan Wang. ## Attributes The date of birth of Shelli Beltran is 0958-03-08. The occupation of Shelli Beltran is occupational therapist. The hobby of Shelli Beltran is sociology.
Thought 3: The child of Shelli Beltran is Aida Wang, Barabara Beltran, Vicki Hackworth. So the answer is Aida Wang, Barabara Beltran, Vicki Hackworth.
Action 3: Finish[Aida Wang{constants.answer_sep}Barabara Beltran{constants.answer_sep}Vicki Hackworth].

Example 5:
Question: Who is the uncle of William Smock?
Thought 1: First, I need to retrieve article about William Smock. To find the uncle of William Smock, I need to find the brother of his parent.
Action 1: RetrieveArticle[William Smock].

Observation 1: # Williams Smock ## Family The parents of Williams Smock are Dominique Smock, Gene Smock.  The mother of Williams Smock is Dominique Smock.  The father of Williams Smock is Gene Smock.  The children of Williams Smock are Shelli Beltran, Stacia Toombs.  The daughters of Williams Smock are Shelli Beltran, Stacia Toombs.  The wife of Williams Smock is Alison Smock.  ## Friends The friend of Williams Smock is Lannie Smock.  ## Attributes The date of birth of Williams Smock is 0926-04-04.  The occupation of Williams Smock is clinical biochemist.  The hobby of Williams Smock is social studies.
Thought 2: The parents of William Smock are Dominique Smock, Gene Smock. To find the uncle of William Smock, I need to find the brother of Dominique Smock and Gene Smock.
Action 2: RetrieveArticle[Dominique Smock].
Observation 2: # Dominique Smock ## Family The child of Dominique Smock is Williams Smock.  The son of Dominique Smock is Williams Smock.  The husband of Dominique Smock is Gene Smock.  ## Friends The friends of Dominique Smock are Gene Smock, Isiah Lutz, Orlando Beltran, Vicki Hackworth.  ## Attributes The date of birth of Dominique Smock is 0897-09-08.  The occupation of Dominique Smock is sports therapist.  The hobby of Dominique Smock is dominoes.
Thought 3: Dominique Smock has no brother. Now I need to retrieve article about Gene Smock and find his brother.
Action 3: RetrieveArticle[Gene Smock].
Observation 3: # Gene Smock ## Family The parents of Gene Smock are Alvaro Smock, Lannie Smock.  Gene Smock's sibling is Eli Smock.  The brother of Gene Smock is Eli Smock.  The mother of Gene Smock is Lannie Smock.  The father of Gene Smock is Alvaro Smock.  The child of Gene Smock is Williams Smock.  The son of Gene Smock is Williams Smock.  The wife of Gene Smock is Dominique Smock.  ## Friends The friends of Gene Smock are Leeann Hackworth, Leisa Lutz, Ricardo Hackworth, Alvaro Smock, Dominique Smock.  ## Attributes The date of birth of Gene Smock is 0898-08-16.  The occupation of Gene Smock is immunologist.  The hobby of Gene Smock is architecture.
Thought 4: The brother of Gene Smock is Eli Smock, and Dominique Smock has no brother. Since Gene Smock and Dominique Smock are the parents of William Smock, the uncle of William Smock is Eli Smock. So the answer is Eli Smock.
Action 4: Finish[Eli Smock].

Example 6:
Question: What is the occupation of the sister of the grandmother of Virgil Hackworth?
Thought 1: First, I need to retrieve article about Virgil Hackworth. To find the grandmother of Virgil Hackworth, I need to find the mother of his parent.
Action 1: RetrieveArticle[Virgil Hackworth].
Observation 1: # Virgil Hackworth ## Family The parents of Virgil Hackworth are Ricardo Hackworth, Vicki Hackworth.  Virgil Hackworth's siblings are Leeann Hackworth, Leisa Lutz.  The sisters of Virgil Hackworth are Leeann Hackworth, Leisa Lutz.  The mother of Virgil Hackworth is Vicki Hackworth.  The father of Virgil Hackworth is Ricardo Hackworth.  ## Friends The friends of Virgil Hackworth are Alison Smock, Eli Smock, Orlando Beltran, Ryan Wang.  ## Attributes The date of birth of Virgil Hackworth is 1009-06-23.  The occupation of Virgil Hackworth is sound technician.  The hobby of Virgil Hackworth is wikipedia editing.
Thought 2: The parents of Virgil Hackworth are Ricardo Hackworth, Vicki Hackworth. To find the grandmother of Virgil Hackworth, I need to find the mother of Ricardo Hackworth and Vicki Hackworth.

Action 2: RetrieveArticle[Ricardo Hackworth].
Observation 2: # Ricardo Hackworth ## Family The children of Ricardo Hackworth are Leeann Hackworth, Leisa Lutz, Virgil Hackworth.  The son of Ricardo Hackworth is Virgil Hackworth.  The daughters of Ricardo Hackworth are Leeann Hackworth, Leisa Lutz.  The wife of Ricardo Hackworth is Vicki Hackworth. ## Friends The friends of Ricardo Hackworth are Gene Smock, Isiah Lutz, Johnetta Wang.  ## Attributes The date of birth of Ricardo Hackworth is 0983-02-24.  The occupation of Ricardo Hackworth is clinical research associate.  The hobby of Ricardo Hackworth is dairy farming.
Thought 3: Ricardo Hackworth has no mother. So I need to retrieve article about Vicki Hackworth and find her mother.
Action 3: RetrieveArticle[Vicki Hackworth].
Observation 3: # Vicki Hackworth ## Family The parents of Vicki Hackworth are Dino Beltran, Shelli Beltran.  Vicki Hackworth's siblings are Aida Wang, Barabara Beltran.  The sisters of Vicki Hackworth are Aida Wang, Barabara Beltran.  The mother of Vicki Hackworth is Shelli Beltran.  The father of Vicki Hackworth is Dino Beltran.  The children of Vicki Hackworth are Leeann Hackworth, Leisa Lutz, Virgil Hackworth.  The son of Vicki Hackworth is Virgil Hackworth.  The daughters of Vicki Hackworth are Leeann Hackworth, Leisa Lutz.  The husband of Vicki Hackworth is Ricardo Hackworth.  ## Friends The friends of Vicki Hackworth are Brian Beltran, Dominique Smock, Eli Smock.  ## Attributes The date of birth of Vicki Hackworth is 0985-05-30.  The occupation of Vicki Hackworth is police officer.  The hobby of Vicki Hackworth is meditation.
Thought 4: The mother of Vicki Hackworth is Shelli Beltran. So the only grandmother of Virgil Hackworth is Shelli Beltran. Now I need to retrieve article about Shelli Beltran and find her sister.
Action 4: RetrieveArticle[Shelli Beltran].
Observation 4: # Shelli Beltran ## Family The parents of Shelli Beltran are Alison Smock, Williams Smock.  Shelli Beltran's sibling is Stacia Toombs.  The sister of Shelli Beltran is Stacia Toombs.  The mother of Shelli Beltran is Alison Smock.  The father of Shelli Beltran is Williams Smock.  The children of Shelli Beltran are Aida Wang, Barabara Beltran, Vicki Hackworth.  The daughters of Shelli Beltran are Aida Wang, Barabara Beltran, Vicki Hackworth.  The husband of Shelli Beltran is Dino Beltran.  ## Friends The friends of Shelli Beltran are Brian Beltran, Eli Smock, Isiah Lutz, Leslee Toombs, Lesley Lutz, Ryan Wang.  ## Attributes The date of birth of Shelli Beltran is 0958-03-08.  The occupation of Shelli Beltran is occupational therapist.  The hobby of Shelli Beltran is sociology.
Thought 5: The sister of Shelli Beltran is Stacia Toombs. Now I need to retrieve article about Stacia Toombs and find her occupation.
Action 5: RetrieveArticle[Stacia Toombs].
Observation 5: # Stacia Toombs ## Family The parents of Stacia Toombs are Alison Smock, Williams Smock.  Stacia Toombs's sibling is Shelli Beltran.  The sister of Stacia Toombs is Shelli Beltran.  The mother of Stacia Toombs is Alison Smock.  The father of Stacia Toombs is Williams Smock.  The child of Stacia Toombs is Leslee Toombs.  The daughter of Stacia Toombs is Leslee Toombs.  The husband of Stacia Toombs is Wilbert Toombs.  ## Friends The friends of Stacia Toombs are Brian Beltran, Isiah Lutz, Leeann Hackworth, Lesley Lutz, Ryan Wang.  ## Attributes The date of birth of Stacia Toombs is 0959-03-22.  The occupation of Stacia Toombs is actuary.  The hobby of Stacia Toombs is finance.
Thought 6: The occupation of Stacia Toombs is actuary. So the answer is actuary.
Action 6: Finish[actuary].

Example 7:
Question: Who is the brother of the person whose occupation is associate
    professor?
Thought 1: First, I need to search for people whose occupation is associate
    professor.
Action 1: Search[associate professor].
Observation 1: (1) Dino Beltran
Thought 2: Person whose hobby is associate professor is Dino Beltran. Now I need
    to retrieve article about Dino Beltran and find his brother.
Action 2: RetrieveArticle[Dino Beltran].
Observation 2: # Dino Beltran ## Family The parents of Dino Beltran are Brian
    Beltran, Daisy Beltran.  Dino Beltran's sibling is Orlando Beltran.  The
    brother of Dino Beltran is Orlando Beltran.  The mother of Dino Beltran is
    Daisy Beltran.  The father of Dino Beltran is Brian Beltran.  The children of
     Dino Beltran are Aida Wang, Barabara Beltran, Vicki Hackworth.  The
    daughters of Dino Beltran are Aida Wang, Barabara Beltran, Vicki Hackworth.
    The wife of Dino Beltran is Shelli Beltran.  ## Friends The friend of Dino
    Beltran is Alvaro Smock.  ## Attributes The date of birth of Dino Beltran is
    0958-08-09.  The occupation of Dino Beltran is associate professor.  The
    hobby of Dino Beltran is shogi.
Thought 3: The brother of Dino Beltran is Orlando Beltran, so the answer is
    Orlando Beltran.
Action 3: Finish[Orlando Beltran].

Example 8:
Question: What is the date of birth of the person whose hobby is meteorology?
Thought 1: First, I need to search for people whose hobby is meteorology.
Action 1: Search[meteorology].
Observation 1: (1) Alison Smock (2) Barabara Beltran
Thought 2: People whose hobby is meteorology are Alison Smock, Barabara Beltran.
    Now I need to retrieve article about Alison Smock and Barabara Beltran and
    find their date of birth.
Action 2: RetrieveArticle[Alison Smock].
Observation 2: # Alison Smock ## Family The children of Alison Smock are Shelli
    Beltran, Stacia Toombs.  The daughters of Alison Smock are Shelli Beltran,
    Stacia Toombs.  The husband of Alison Smock is Williams Smock.  ## Friends
    The friends of Alison Smock are Eli Smock, Lannie Smock, Virgil Hackworth.
    ## Attributes The date of birth of Alison Smock is 0929-10-28.  The
    occupation of Alison Smock is health promotion specialist.  The hobby of
    Alison Smock is meteorology.
Thought 3: The date of birth of Alison Smock is 0929-10-28. Now I need to
    retrieve article about Barabara Beltran and find her date of birth.
Action 3: RetrieveArticle[Barabara Beltran].
Observation 3: # Barabara Beltran ## Family The parents of Barabara Beltran are
    Dino Beltran, Shelli Beltran.  Barabara Beltran's siblings are Aida Wang,
    Vicki Hackworth.  The sisters of Barabara Beltran are Aida Wang, Vicki
    Hackworth.  The mother of Barabara Beltran is Shelli Beltran.  The father of
    Barabara Beltran is Dino Beltran.  ## Friends ## Attributes The date of birth
     of Barabara Beltran is 0989-06-11.  The occupation of Barabara Beltran is
    broadcast engineer.  The hobby of Barabara Beltran is meteorology.
Thought 4: The date of birth of Barabara Beltran is 0989-06-11. So the answer is
    0929-10-28, 0989-06-11.
Action 4: Finish[0929-10-28{constants.answer_sep}0989-06-11].

Example 9:

Question: Who is the cousin of the person whose occupation is broadcast engineer?

Thought 1: First, I need to search for people whose occupation is broadcast engineer.

Action 1: Search[broadcast engineer].

Observation 1: # Barabara Beltran ## Family The parents of Barabara Beltran are Dino Beltran, Shelli Beltran.  Barabara Beltran's siblings are Aida Wang, Vicki Hackworth.  The sisters of Barabara Beltran are Aida Wang, Vicki Hackworth.  The mother of Barabara Beltran is Shelli Beltran.  The father of Barabara Beltran is Dino Beltran.  ## Friends ## Attributes The date of birth of Barabara Beltran is 0989-06-11.  The occupation of Barabara Beltran is broadcast engineer.  The hobby of Barabara Beltran is meteorology.

Thought 2: Person whose hobby is broadcast engineer is Barabara Beltran. The cousin of Barabara Beltran is the child of sibling of her parent. Barabara Beltran's parents are Dino Beltran and Shelly Beltran. Now I need to find the sibling of Dino Beltran and Shelly Beltran.

Action 2: RetrieveArticle[Dino Beltran].

Observation 2: # Dino Beltran ## Family The parents of Dino Beltran are Brian Beltran, Daisy Beltran.  Dino Beltran's sibling is Orlando Beltran.  The brother of Dino Beltran is Orlando Beltran.  The mother of Dino Beltran is Daisy Beltran.  The father of Dino Beltran is Brian Beltran.  The children of Dino Beltran are Aida Wang, Barabara Beltran, Vicki Hackworth.  The daughters of Dino Beltran are Aida Wang, Barabara Beltran, Vicki Hackworth.  The wife of Dino Beltran is Shelli Beltran.  ## Friends The friend of Dino Beltran is Alvaro Smock.  ## Attributes The date of birth of Dino Beltran is 0958-08-09.  The occupation of Dino Beltran is associate professor.  The hobby of Dino Beltran is shogi.

Thought 3: The sibling of Dino Beltran is Orlando Beltran. Now I need to retrieve article about Orlando Beltran and find his child.

Action 3: RetrieveArticle[Orlando Beltran].

Observation 3: # Orlando Beltran ## Family The parents of Orlando Beltran are Brian Beltran, Daisy Beltran.  Orlando Beltran's sibling is Dino Beltran.  The brother of Orlando Beltran is Dino Beltran.  The mother of Orlando Beltran is Daisy Beltran.  The father of Orlando Beltran is Brian Beltran.  ## Friends The friends of Orlando Beltran are Virgil Hackworth, Dominique Smock.  ## Attributes The date of birth of Orlando Beltran is 0953-10-23.  The occupation of Orlando Beltran is petroleum engineer.  The hobby of Orlando Beltran is learning.

Thought 4: Orlando Beltran has no child, so no cousin from the side of Dino Beltran. Now I need to retrieve article about Shelli Beltran and find her sibling.

Action 4: RetrieveArticle[Shelli Beltran].

Observation 4: # Shelli Beltran ## Family The parents of Shelli Beltran are Alison Smock, Williams Smock.  Shelli Beltran's sibling is Stacia Toombs.  The sister of Shelli Beltran is Stacia Toombs.  The mother of Shelli Beltran is Alison Smock.  The father of Shelli Beltran is Williams Smock.  The children of Shelli Beltran are Aida Wang, Barabara Beltran, Vicki Hackworth.  The daughters of Shelli Beltran are Aida Wang, Barabara Beltran, Vicki Hackworth.  The husband of Shelli Beltran is Dino Beltran.  ## Friends The friends of Shelli Beltran are Brian Beltran, Eli Smock, Isiah Lutz, Leslee Toombs, Lesley Lutz, Ryan Wang.  ## Attributes The date of birth of Shelli Beltran is 0958-03-08.  The occupation of Shelli Beltran is occupational therapist.  The hobby of Shelli Beltran is sociology.

Thought 5: The sibling of Shelli Beltran is Stacia Toombs. Now I need to retrieve
    article about Stacia Toombs and find her child, to get the cousin of
    Barabara Beltran.
Action 5: RetrieveArticle[Stacia Toombs].
Observation 5: # Stacia Toombs ## Family The parents of Stacia Toombs are Alison
    Smock, Williams Smock.  Stacia Toombs's sibling is Shelli Beltran.  The
    sister of Stacia Toombs is Shelli Beltran.  The mother of Stacia Toombs is
    Alison Smock.  The father of Stacia Toombs is Williams Smock.  The child of
    Stacia Toombs is Leslee Toombs.  The daughter of Stacia Toombs is Leslee
    Toombs.  The husband of Stacia Toombs is Wilbert Toombs.  ## Friends The
    friends of Stacia Toombs are Brian Beltran, Isiah Lutz, Leeann Hackworth,
    Lesley Lutz, Ryan Wang.  ## Attributes The date of birth of Stacia Toombs is
    0959-03-22.  The occupation of Stacia Toombs is actuary.  The hobby of Stacia
     Toombs is finance.
Thought 6: The child of Stacia Toombs is Leslee Toombs. So the cousin of Barabara
     Beltran is Leslee Toombs. The answer is Leslee Toombs.
Action 6: Finish[Leslee Toombs].

Example 10:
Question: Who is the great-granddaughter of the person whose hobby is biology?
Thought 1: First, I need to search for people whose hobby is biology.
Action 1: Search[biology].
Observation 1: (1) Alvaro Smock
Thought 2: Person whose hobby is biology is Alvaro Smock. Now I need to retrieve
    article about Alvaro Smock and find his great-granddaughter. A great-
    granddaughter is the daughter of the child of the child of the person.
Action 2: RetrieveArticle[Alvaro Smock].
Observation 2: # Alvaro Smock ## Family The children of Alvaro Smock are Eli
    Smock, Gene Smock.  The sons of Alvaro Smock are Eli Smock, Gene Smock.  The
    wife of Alvaro Smock is Lannie Smock.  ## Friends The friends of Alvaro Smock
     are Dino Beltran, Gene Smock, Aida Wang.  ## Attributes The date of birth of
     Alvaro Smock is 0867-07-12.  The occupation of Alvaro Smock is osteopath.
    The hobby of Alvaro Smock is biology.
Thought 3: The children of Alvaro Smock are Eli Smock, Gene Smock. Now I need to
    retrieve article about Eli Smock and Gene Smock and find their child.
Action 3: RetrieveArticle[Eli Smock].
Observation 3: # Eli Smock ## Family The parents of Eli Smock are Alvaro Smock,
    Lannie Smock.  Eli Smock's sibling is Gene Smock.  The brother of Eli Smock
    is Gene Smock.  The mother of Eli Smock is Lannie Smock.  The father of Eli
    Smock is Alvaro Smock.  ## Friends The friends of Eli Smock are Leisa Lutz,
    Shelli Beltran, Vicki Hackworth, Virgil Hackworth, Alison Smock, Brian
    Beltran.  ## Attributes The date of birth of Eli Smock is 0901-01-18.  The
    occupation of Eli Smock is retail manager.  The hobby of Eli Smock is tether
    car.
Thought 4: Eli Smock has no child. Now I need to retrieve article about Gene
    Smock and find his child.
Action 4: RetrieveArticle[Gene Smock].
Observation 4: # Gene Smock ## Family The parents of Gene Smock are Alvaro Smock,
     Lannie Smock.  Gene Smock's sibling is Eli Smock.  The brother of Gene Smock
     is Eli Smock.  The mother of Gene Smock is Lannie Smock.  The father of Gene
     Smock is Alvaro Smock.  The child of Gene Smock is Williams Smock.  The son
    of Gene Smock is Williams Smock.  The wife of Gene Smock is Dominique Smock.
     ## Friends The friends of Gene Smock are Leeann Hackworth, Leisa Lutz,
    Ricardo Hackworth, Alvaro Smock, Dominique Smock.  ## Attributes The date of

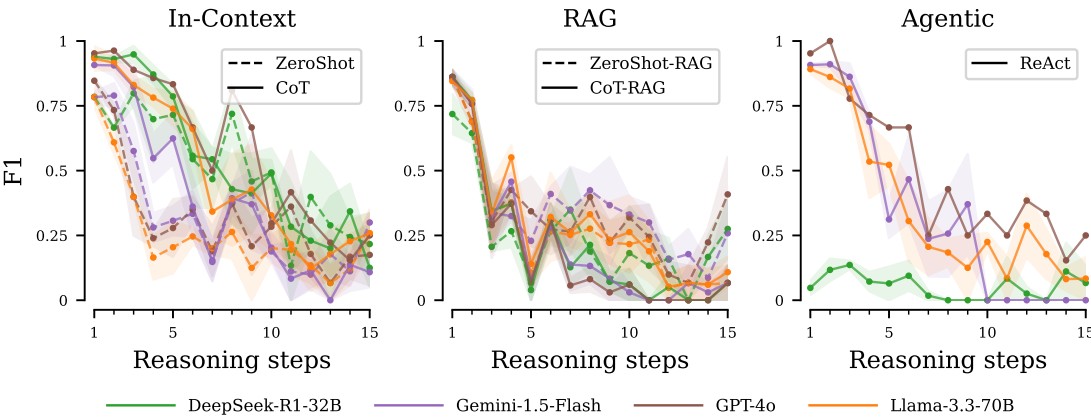

*Figure 7.* **F1 scores versus question difficulty, measured by *reasoning steps*, for questions with exactly 1 solution.** We observe similar trends as in Figure 3, demonstrating that the number of solutions is not solely responsible for the drop in performance.

```
    birth of Gene Smock is 0898-08-16.  The occupation of Gene Smock is
    immunologist.  The hobby of Gene Smock is architecture.
Thought 5: The child of Gene Smock is Williams Smock. Now I need to retrieve
    article about Williams Smock and find his daughter, to get the great-
    granddaughter of Alvaro Smock.
Action 5: RetrieveArticle[Williams Smock].
Observation 5: # Williams Smock ## Family The parents of Williams Smock are
    Dominique Smock, Gene Smock.  The mother of Williams Smock is Dominique Smock
    .  The father of Williams Smock is Gene Smock.  The children of Williams
    Smock are Shelli Beltran, Stacia Toombs.  The daughters of Williams Smock are
     Shelli Beltran, Stacia Toombs.  The wife of Williams Smock is Alison Smock.
     ## Friends The friend of Williams Smock is Lannie Smock.  ## Attributes The
    date of birth of Williams Smock is 0926-04-04.  The occupation of Williams
    Smock is clinical biochemist.  The hobby of Williams Smock is social studies.
Thought 6: The daughters of Williams Smock are Shelli Beltran, Stacia Toombs. So
    the great-granddaughters of Alvaro Smock, whose hobby is biology, are Shelli
    Beltran, Stacia Toombs. The answer is Shelli Beltran, Stacia Toombs.
Action 6: Finish[Shelli Beltran{constants.answer_sep}Stacia Toombs].
(END OF EXAMPLES)

Now answer the following question:
Question: {{question}}
{{scratchpad}}
```

## D. Additional Results

### D.1. Multi-Hop RAG Baselines

We include two RAG baselines that interleave reasoning with retrieval: Self-Ask (Press et al., 2023) and IRCoT (Trivedi et al., 2023). We use the implementation from FlashRAG (https://github.com/RUC-NLPIR/FlashRAG) and write few-shot examples suited to PhantomWiki. Table 4 includes the results of these baselines on the same PhantomWiki instances as in Table 2. Notably, IRCoT improves over ZEROSHOT-RAG and COT-RAG, but struggles on questions requiring many reasoning steps, as revealed by Figure 8.

*Table 4.* **F1 scores (in %) for Llama-3.3-70B with RAG prompting.** ZEROSHOT-RAG and COT-RAG use the UAE-LARGE-V1 retriever model (top-k= 4); SELF-ASK and IRCOT use the BM25 retriever (top-k= 5). We report mean $\pm$ 1 standard error across 3 dataset generation seeds.

| Universe Size $n$ | ZEROSHOT-RAG | COT-RAG | SELF-ASK | IRCOT |
|---|---|---|---|---|
| 50 | 17.55 ± 2.20 | 20.01 ± 1.81 | 15.77 ± 3.22 | 23.93 ± 0.93 |

## D.2. Article Rephrasing

To explore how LLMs can be used to improve the realism of our templated articles, we instruct Llama-3.3-70B to rephrase articles in corpora of size $n = 50$ from Figure 3. Our first, "short" prompt instructs the LLM to condense the templated articles, while still retaining all factual information:

```
Shuffle and rephrase the following wikipedia-like article. Keep ALL facts
like name, relation, date, occupation, hobby and gender exactly as stated. Do
not add ANY new information.
```

Our second, "long" prompt permits the LLM to introduce new facts, without contradicting existing facts:

```
Transform the following factual article into an engaging narrative profile:

1. Add colorful descriptions and personality traits that might be inferred
2. Create a vivid backstory about how relationships formed
3. Elaborate on the person's career path and achievements
4. Describe their hobbies in rich detail, including when they might have started
    them
5. Include hypothetical quotes from family members
6. Imagine and describe the person's daily routine
7. Add details about where they might live and their home environment
8. Write in a warm, personal tone as if you've known the subject for years
9. Maintain a natural flow

When transforming the article, ensure the following:
1. All factual information remains 100% accurate (names, relationships, dates,
```

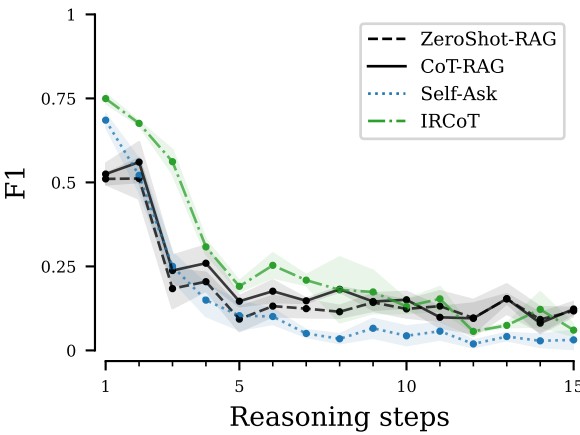

*Figure 8.* **F1 scores versus question difficulty, measured by reasoning steps.** ZEROSHOT-RAG and COT-RAG use the UAE-LARGE-V1 retriever model (top-k= 4); SELF-ASK and IRCOT use the BM25 retriever (top-k= 5). For all methods, we use Llama-3.3-70B as the generator model. We report mean $\pm$ 1 standard error across 3 dataset generations seeds with universe size $n = 50$.

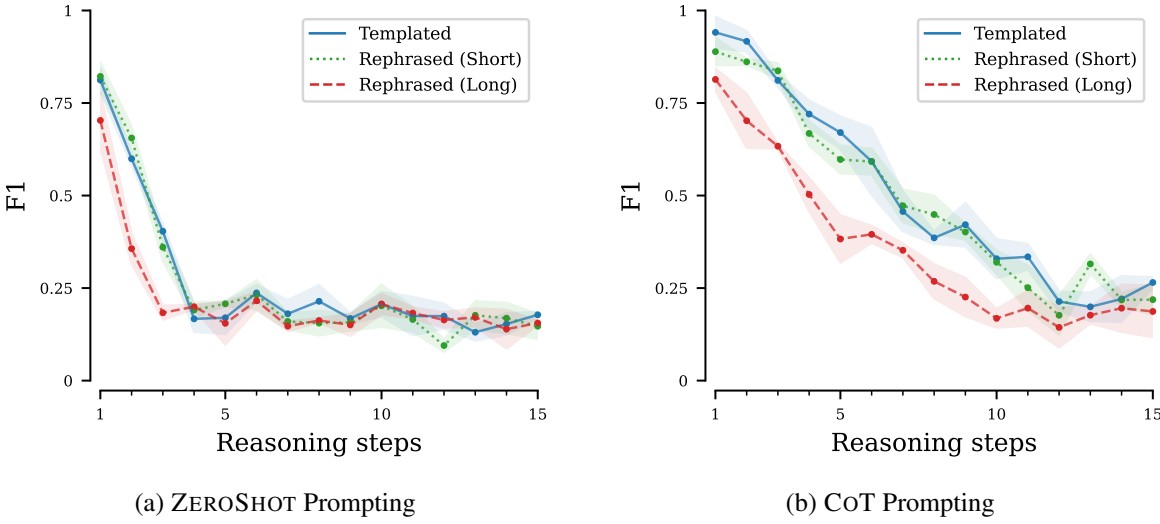

(a) ZEROSHOT Prompting          (b) COT Prompting

*Figure 9.* **F1 scores versus question difficulty, measured by reasoning steps, for various article styles**. We rephrase templated articles generated by PhantomWiki with Llama-3.3-70B. Then for evaluation, we replace templated articles with LLM-rephrased articles in ZEROSHOT and COT prompts from Appendix C.1 and Appendix C.3 respectively. As in Figure 3, we report mean $\pm$ 1 standard error across 3 dataset generation seeds for universe size $n = 50$. We use evaluation sampling hyper-parameters from Section 4.

```
   occupation, hobby, gender)
2. No new information is added or implied
3. The content is presented in a natural, flowing narrative rather than just
   shuffling bullet points
4. Different sections are reorganized in a way that still makes logical sense
5. Sentence structures and vocabulary are varied from the original

This is the article to transform:
```

To generate articles of either styles, we prompt Llama-3.3-70B with temperature $= 0.7$, top-p $= 0.9$, and max output tokens $= 1024$. We include an example short article in Appendix D.2.1 and an example long article in Appendix D.2.3. Figure 9 shows plots of F1 scores vs question difficulty of Llama-3.3-70B using ZEROSHOT and COT prompting on these LLM-rephrased articles. We find that regardless of the article style—templated or LLM-rephrased—F1 scores decline as the number of reasoning steps increases.

### D.2.1. EXAMPLE OF TEMPLATED ARTICLE

```
# Alison Smock

## Family
The daughters of Alison Smock are Cythia Smock, Shelli Beltran, Stacia Toombs.
The husband of Alison Smock is Williams Smock.

## Friends
The friends of Alison Smock are Cortney Parmer, Jamison Baptiste, Wilbert Toombs.

## Attributes
The date of birth of Alison Smock is 0929-10-28.
The occupation of Alison Smock is broadcast engineer.
The hobby of Alison Smock is meteorology.
The gender of Alison Smock is female.
```

D.2.2. EXAMPLE OF SHORT REPHRASED ARTICLE

The following article was generated with Llama-3.3-70B using 118 output tokens.

> Alison Smock, a female born on 0929-10-28, is a broadcast engineer with a hobby of meteorology. She is married to Williams Smock and they have daughters named Cythia Smock, Shelli Beltran, and Stacia Toombs. Alison Smock's social circle includes friends such as Cortney Parmer, Jamison Baptiste, and Wilbert Toombs.

D.2.3. EXAMPLE OF LONG REPHRASED ARTICLE

The following article was generated with Llama-3.3-70B using 669 output tokens. Llama-3.3-70B incorrectly stated the date-of-birth (red) and omitted the last names of the friends (orange).

> Alison Smock, a brilliant and charismatic broadcast engineer, has been fascinated by the intricacies of technology and the wonders of the atmosphere for as long as anyone can remember. Born on October 28, 1929, Alison grew up with an insatiable curiosity and a passion for learning that would serve her well throughout her life. As a young girl, she was always tinkering with gadgets and machinery, trying to understand how they worked, and this innate curiosity eventually led her to pursue a career in broadcast engineering.
>
> As she navigated her professional path, Alison met her husband, Williams Smock, and together they built a loving family. They had three beautiful daughters, Cythia, Shelli, and Stacia, who brought immense joy and fulfillment to their lives. According to her daughter Cythia, "Mom was always there for us, supporting us and encouraging us to pursue our dreams, no matter how big or small they seemed." Alison's family was her rock, and she cherished every moment she spent with them.
>
> Alison's career as a broadcast engineer was marked by numerous achievements and accomplishments. She was known for her exceptional problem-solving skills and her ability to stay calm under pressure. Her colleagues admired her dedication and work ethic, and she quickly became an invaluable member of her team. When she wasn't working, Alison loved to indulge in her hobby of meteorology. She found the study of weather patterns and atmospheric conditions absolutely captivating, and she spent hours poring over books and watching documentaries on the subject. As her friend Cortney Parmer recalls, "Alison would often call me up and say, 'Did you see that storm system moving in? I've been tracking it all day!' She was like a kid in a candy store when it came to weather."
>
> In her free time, Alison enjoyed spending time with her friends, including Cortney, Jamison Baptiste, and Wilbert Toombs. They would often meet up for coffee or go on hikes together, discussing everything from the latest news to their personal lives. Wilbert, who is also the husband of Alison's daughter Stacia, says, "Alison is like a mother to me. She's always been so kind and supportive, and I admire her intelligence and passion for life."
>
> A typical day for Alison begins early, with a quiet moment of reflection and a cup of coffee on her back porch, surrounded by the beauty of nature. She lives in a cozy home filled with books, gadgets, and mementos from her travels, and she loves to spend time in her garden, tending to her plants and watching the birds. As she gets ready for work, Alison feels grateful for the life she's built and the people who bring joy and love into her world. As her husband Williams says, "Alison is an amazing woman, and I feel so lucky to have her by my side. She's intelligent, kind, and beautiful, and I admire her every day." With her quick wit, sharp mind, and generous spirit, Alison Smock is a true treasure, and those who know her feel fortunate to have her in their lives.

**D.3. Fine-tuning on PhantomWiki– Experiment Details**

PhantomWiki creates datasets with relationships, names, attributes populated randomly on-demand, ensuring that each dataset instance is unique and robust to LLMs memorizing dataset facts. Even so, it is possible that fine-tuning on PhantomWiki-generated datasets could improve performance by leveraging language and template structure in PhantomWiki articles and questions. Accordingly, we fine-tune LLMs on PhantomWiki datasets to (1) assess the viability of PhantomWiki for training LLMs and (2) test the robustness of PhantomWiki on-demand datasets from memorization.

We generate 10 new PhantomWiki dataset instances (question depth 20 and universe size 50) amounting to 5K training question-answer pairs. We then perform full fine-tuning of Qwen2.5-0.5B-Instruct (Qwen Team, 2024) and parameter-efficient fine-tuning of Qwen2.5-3B-Instruct with LoRA (Hu et al., 2022) applied to all linear layers. We use a node of 4

A100 GPUs each with 80GB GPU VRAM for each experiment. For each base model, we employ two popular fine-tuning algorithms:

1. The first is **Group Relative Policy Optimization (GRPO)** (DeepSeek AI, 2024), where we elicit chain-of-thought response from the LLM using our CoT template (Appendix C.3) and return reward as the F1-score of the generated answer list against the ground-truth answer list.

   We use the GRPO implementation from Huggingface's TRL library. We set max prompt length $= 4096$, sufficient to include our prompt of the 50 articles of the universe, and limit the max completion length to $= 128$. We reserve 1 GPU for generation with vLLM, and use the other 3 GPUs for training to get 6 total generations per step of GRPO update. For full fine-tuning, we set per-device training batch size $= 8$ and gradient accumulation steps $= 8$, and for LoRA fine-tuning we set 2 and 4 respectively. We fine-tune for 3 epochs using the AdamW optimizer with initial learning rate set to $5 \times 10^{-6}$ for full fine-tuning and $10^{-4}$ for LoRA fine-tuning.

2. The second is **Supervised Fine-tuning (SFT)**, where we provide all articles and the question using our ZEROSHOT prompt in Appendix C.1 and train on the ground-truth answer list.

   Again we use the SFT implementation from Huggingface's TRL library and set the max prompt length to $4096$. For both full fine-tuning and LoRA fine-tuning, we set per-device training batch size and gradient accumulation steps to $1$. We fine-tune for 3 epochs using the AdamW optimizer with initial learning rate set to $2 \times 10^{-5}$ for full fine-tuning and $10^{-4}$ for LoRA fine-tuning.

We evaluate these models on the three PhantomWiki dataset instances of size $n = 50$ and maximum recursion depth 20 (500 questions per dataset instance), the same used in Table 2 and Figure 3. Figure 10 shows F1 scores as a function of reasoning steps, comparing base LLMs with fine-tuned ones, and Table 3 shows the aggregate F1 scores. To adhere to the fine-tuning experiment setup, SFT-trained LLMs are evaluated with ZEROSHOT prompt, and GRPO-trained with CoT prompt.

For both full fine-tuned Qwen2.5-0.5B-Instruct and LoRA fine-tuned Qwen2.5-3B-Instruct, we see that GRPO improves performance for all question difficulties compared to CoT and ZEROSHOT prompting. We find that SFT does not improve performance over ZEROSHOT prompting. Notably, performance of fine-tuned LLMs is still far from optimal and the F1 scores struggle as question difficulty increases.

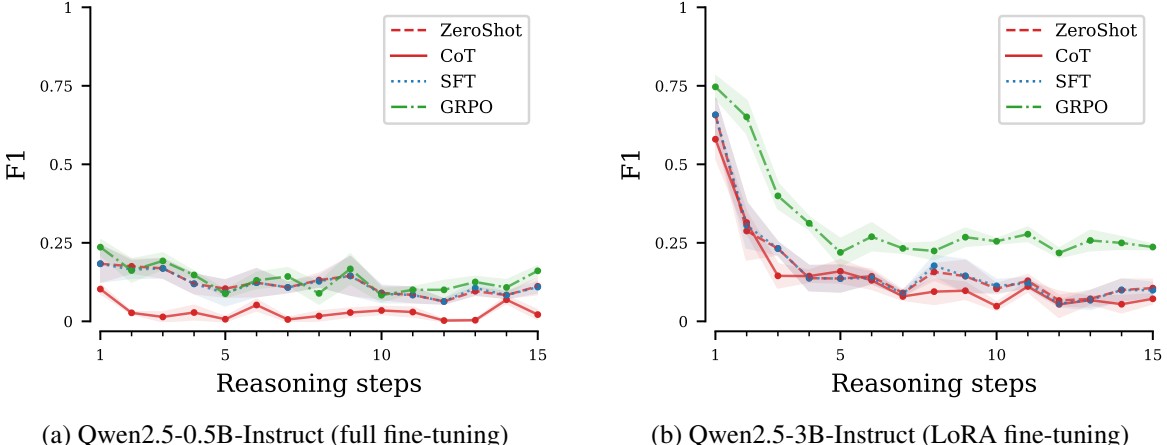

(a) Qwen2.5-0.5B-Instruct (full fine-tuning)      (b) Qwen2.5-3B-Instruct (LoRA fine-tuning)

*Figure 10.* **F1 scores versus question difficulty, measured by reasoning steps.** We report mean $\pm$ 1 standard error across 3 dataset generation seeds for universe size $n = 50$. ZEROSHOT and CoT are prompting methods; SFT and GRPO are fine-tuning methods.

## E. Example of Small Corpus

We generate a universe of size $n = 4$, setting the number of family trees to be one. We include the articles below.

### E.1. Article 1

```
# Claud Colin

## Family
The brother of Claud Colin is Mckinley Colin.
The mother of Claud Colin is Ramona Colin.
The father of Claud Colin is Danilo Colin.

## Friends
The friend of Claud Colin is Danilo Colin.

## Attributes
The date of birth of Claud Colin is 0241-12-06.
The occupation of Claud Colin is academic librarian.
The hobby of Claud Colin is amateur astronomy.
The gender of Claud Colin is male.
```

### E.2. Article 2

```
# Danilo Colin

## Family
The sons of Danilo Colin are Claud Colin, Mckinley Colin.
The wife of Danilo Colin is Ramona Colin.

## Friends
The friends of Danilo Colin are Mckinley Colin, Ramona Colin, Claud Colin.

## Attributes
The date of birth of Danilo Colin is 0219-08-09.
The occupation of Danilo Colin is clinical research associate.
The hobby of Danilo Colin is crystals.
The gender of Danilo Colin is male.
```

**E.3. Article 3**

```
# Mckinley Colin

## Family
The brother of Mckinley Colin is Claud Colin.
The mother of Mckinley Colin is Ramona Colin.
The father of Mckinley Colin is Danilo Colin.

## Friends
The friends of Mckinley Colin are Ramona Colin, Danilo Colin.

## Attributes
The date of birth of Mckinley Colin is 0246-10-18.
The occupation of Mckinley Colin is museum curator.
The hobby of Mckinley Colin is stamp collecting.
The gender of Mckinley Colin is male.
```

**E.4. Article 4**

```
# Ramona Colin

## Family
The sons of Ramona Colin are Claud Colin, Mckinley Colin.
The husband of Ramona Colin is Danilo Colin.

## Friends
The friends of Ramona Colin are Danilo Colin, Mckinley Colin.

## Attributes
The date of birth of Ramona Colin is 0219-09-08.
The occupation of Ramona Colin is technical sales engineer.
The hobby of Ramona Colin is trainspotting.
The gender of Ramona Colin is female.
```

