# OpenReview forum: "PhantomWiki: On-Demand Datasets for Reasoning and Retrieval Evaluation"
_ICML.cc/2025/Conference — ICML 2025 poster_

### Official Review · Reviewer_w1Q5 · 2025-03-11

**Overall Recommendation:** 3

**Summary:**

The paper proposes PhantomWiki, a benchmark that dynamically generates a fictional universe for evaluating retrieval and multi-hop reasoning. The proposed benchmark can be generated on-demand and is free of data leakage because the fictional events are independent with the real-world.

The PhantomWiki data generation pipeline first generates a random universe of n characters, as well as their social relationships and personal facts, e.g., date of birth, job, and hobby. The facts are then converted into articles by filling pre-defined templates. The questions are also generated by templates and answers derived using Prolog by inferring from the rules of the universe.

Experiments are carried out with varying difficulty levels and universe sizes: 1/ in-context prompting that the entire universe is placed in the context window, 2/ RAG prompting where a neural retriever is used to search for top-4 most relevant documents, 3/ Agentic prompting where tool use is enabled.

**Claims And Evidence:**

This is a dataset paper. I think the main claim could be viewed as "PhantomWiki is a scalable and data leakage-resistant framework for disentangled evaluation of reasoning, retrieval, and tool-use abilities." I think this claim is supported because the dataset is generated from a fictional and controlled environment.

**Essential References Not Discussed:**

This is another related work that evaluates reasoning using synthetic data.

Levy, M., Jacoby, A., & Goldberg, Y. (2024, August). Same Task, More Tokens: the Impact of Input Length on the Reasoning Performance of Large Language Models. In Proceedings of the 62nd Annual Meeting of the Association for Computational Linguistics (Volume 1: Long Papers) (pp. 15339-15353).

**Experimental Designs Or Analyses:**

The experiments are technically sound. My main concern is on bias in retrieval evaluation:

The dataset is not well-suited for evaluating retrieval. Although the paper claims to evaluate retrieval, the questions and articles are not appropriate for neural retrieval. For instance, a question like "What is the job of the father of the uncle of the sister of {personA}?" requires iterative bridge entity resolution, making neural retrieval ineffective. In addition, in the retrieval experiment, RAG prompting only retrieved 4 most relevant documents, which are likely insufficient for many multi-hop questions.

**Methods And Evaluation Criteria:**

### Strength
Overall, I liked the considerate design of the PhantomWiki data generation pipeline.
1. It generates a knowledge graph of a universe and uses logical programming to provide answers to multihop questions.
2. The dataset is fictional and free of data leakage.

### Weakness
Article content and questions lack realism. The articles are generated using templates, are very short and lack realism for real-world retrieval augmented generation applications. The multi-hop questions are highly contrived; they are unlikely to appears in real-world situations.

**Other Comments Or Suggestions:**

It would be helpful to define how F1 is calculated in the appendix.

**Other Strengths And Weaknesses:**

No. Please refer to my comments in the above sections.

**Questions For Authors:**

1. Is there any mechanism to resolve conflicts or ambiguous questions? Does all templates lead to answerable questions?
2. If the dataset is focused on context free grammar, is it easy to learn after fine-tuning on the same distribution because CFG is well defined?
3. Is this dataset intended for any training or evaluation only?

**Relation To Broader Scientific Literature:**

Prior work on RAG evaluation often confounds with parametric knowledge. This work create fictional world that decouples the impact of data leakage.

Prior work on reasoning with synthetic data does not consider the retrieval setup where the content may not fit in context window as task complexity grows.

**Theoretical Claims:**

The paper does not make theoretical claims.

---

> ### Author Rebuttal · Authors · 2025-04-01
>
> We thank the reviewer for their positive feedback on the design of PhantomWiki and for acknowledging its novelty in providing a scalable and data leakage-resistant evaluation framework. We address your concerns in turn.
>
> **1. Using templated articles**
>
> > Article content and questions lack realism. The articles are generated using templates, are very short and lack realism for real-world retrieval augmented generation applications.
>
> We agree that the current articles and questions are intentionally stylized and minimalistic due to their templated construction. To explore how LLMs can be leveraged to improve the realism of PhantomWiki articles, we **add new experiments using Llama-3.3-70B-Instruct to rephrase the articles**. We prompt the model in two ways (see https://imgur.com/a/ffvugvg): the “short” prompt instructs the LLM to paraphrase the templated articles, while still retaining all factual information; the “long” prompt permits the LLM to expand on the articles, without contradicting existing facts. We experimented with multiple temperature and top-p settings to mitigate hallucinations while encouraging creative outputs (see e.g., https://imgur.com/a/wFvjAFP).
>
> Additionally, we quantify the effect of using these rephrased articles by reporting F1 scores of Llama-3.3-70B-Instruct with ZeroShot and CoT prompting: see https://imgur.com/a/DQuvQIA.
> Remarkably, we find **similar trends in performance regardless of whether we use LLM-generated or templated articles**. Thus, **templated articles** provide **four key benefits**: (1) they are cheap—no GPU or API costs, (2) fast—no latency from querying LLMs, (3) 100% factually consistent and (4) they allow for larger universe sizes in limited context windows. We leave as future work how to ensure that question-answer pairs are consistent with rephrased articles without relying on human intervention.
>
> **2. Use case of PhantomWiki and evaluating LLMs fine-tuned on PhantomWiki data**
>
> > If the dataset is focused on context free grammar, is it easy to learn after fine-tuning on the same distribution because CFG is well defined?
>
> > Is this dataset intended for any training or evaluation only?
>
> These are great questions! To assess the viability of PhantomWiki for training language models, we **add new fine-tuning experiments**: see https://imgur.com/a/QHftpCM. Specifically, we perform full fine-tuning of Qwen2.5-0.5B-Instruct and parameter-efficient fine-tuning of Qwen2.5-3B-Instruct (via LoRA applied to all linear layers). We also experiment with two popular training algorithms: Group Relative Policy Optimization (GRPO) and supervised fine-tuning (SFT). For Qwen2.5-0.5B, we find that GRPO and SFT both improve F1 compared to prompting-based methods, likely due to improved ability to output the proper answer format. For Qwen2.5-3B, we find that GRPO improves F1 slightly, whereas SFT worsens F1, likely due to overfitting on the training samples. These experiments **show that further advances beyond fine-tuning are needed to truly close the gap on PhantomWiki**. Please see also our rebuttal to reviewer YauR for full experiment details.
>
> **3. Evaluating retrieval**
>
> > the questions and articles are not appropriate for neural retrieval.
>
> > a question like "What is the job of the father of the uncle of the sister of {personA}?" requires iterative bridge entity resolution, making neural retrieval ineffective.
>
> > RAG prompting only retrieved 4 most relevant documents, which are likely insufficient for many multi-hop questions.
>
> To address all three concerns, we **include new multi-hop RAG baselines** (namely, IRCoT and Self-Ask) using BM25 (top-k=5) as the retriever and Llama-3.3-70B-Instruct as the generator model: see https://imgur.com/a/OMjtf3t. We find that these baselines **outperform** the original ZeroShot-RAG and CoT-RAG baselines on **low-difficulty** questions, but **similarly struggle** on questions with **high difficulty**.
>
> **4. Ambiguous questions & Calculation of F1 score**
>
> PhantomWiki questions always have at least one correct answer and often have multiple correct answers, e.g., “Who is the friend of X?” if X has several friends, or “What is the hobby of the sister of X?” if X has multiple sisters. Furthermore, we explicitly instruct the model to return all possible answers (see prompts in App. C). We compare the LLMs’ predicted answer list against the ground-truth list to compute precision, recall, and F1 per question. Final F1 scores (e.g., in Table 2) are averages over all 500 questions in a PhantomWiki instance. **We will include this clarification in Section 4.3.**
>
> **5. Additional references**
>
> We will include your suggested references on synthetic data generation in our final manuscript.
>
> We hope that these **new experiments** and clarifications have significantly strengthened our message and empirical results, and if so we would like to politely ask you to consider raising your score. Thank you!

---

### Official Review · Reviewer_GF7T · 2025-03-13

**Overall Recommendation:** 3

**Summary:**

This paper presents a solution for creating a high-quality benchmark to evaluate the RAG and reasoning abilities of LLMs. Specifically, the proposed method, PhantomWiki, introduces a novel pipeline for generating unique, factually consistent document corpora with diverse question-answer pairs for evaluation. PhantomWiki generates a new instance on demand for each evaluation, effectively mitigating data leakage and inflated performance issues. By varying the difficulty of questions and the size of the corpus, the framework disentangles reasoning from retrieval capabilities.

Through experiments involving various document sizes, question difficulties, and frontier long-context LLMs, PhantomWiki demonstrates itself as a challenging benchmark for state-of-the-art models. It offers a scalable and leakage-resistant evaluation method for assessing reasoning, retrieval, and tool-use abilities.

**Claims And Evidence:**

Please see the “Other Strengths And Weaknesses” section below.

**Essential References Not Discussed:**

N/A

**Experimental Designs Or Analyses:**

Please see the “Other Strengths And Weaknesses” section below.

**Methods And Evaluation Criteria:**

Please see the “Other Strengths And Weaknesses” section below.

**Other Comments Or Suggestions:**

Please see the “Other Strengths And Weaknesses” section.

**Other Strengths And Weaknesses:**

**Strengths:**

1. The proposed benchmark framework, PhantomWiki, is free from data leakage and can be adjusted to control reasoning complexity and retrieval difficulty. The authors also ensure that the generated documents are accurate and error-free, making the dataset high quality.
2. The details of the dataset construction are well-documented, accompanied by solid experiments with various complexity and context lengths, and analysis for evaluating reasoning and retrieval abilities under different conditions.
3. If publicly released, the benchmark framework could benefit the community in evaluating future generations of language models.

**Weaknesses:**

1. This paper could be improved by incorporating more recent baselines. The reason RAG performs poorly in Figure 3 is that it retrieves documents only once. However, several recent methods ([1][2][3], etc.) allow for multiple retrievals, significantly enhancing answer accuracy.
2. While using a context-free grammar and Prolog to generate question-answer pairs is intriguing, the concept of generating questions beyond the pre-training set of LLMs is not novel. For instance, [4][5][6] also generate questions that are resistant to data leakage and are dynamic rather than fixed. The innovation of this paper lies primarily in proposing an unreal universe setting and the ability to dynamically generate questions and corpus sizes of varying difficulty, which may be limited.
3. The domain of the generated questions is limited to the universe. While this helps prevent data leakage, it also restricts the comprehensive evaluation of LLMs. The generated questions tend to be less diverse, primarily focusing on reasoning between personal relationships and objects. In contrast, the questions in [4] are more diverse, covering temporal, spatial, mathematical, social, scientific, and personal contexts.

[1] Enhancing Retrieval-Augmented Large Language Models with Iterative Retrieval-Generation Synergy

[2] Interleaving Retrieval with Chain-of-Thought Reasoning for Knowledge-Intensive Multi-Step Questions

[3] Adaptive-RAG: Learning to Adapt Retrieval-Augmented Large Language Models through Question Complexity

[4] ToolQA: A Dataset for LLM Question Answering with External Tools

[5] RealTime QA: What's the Answer Right Now?

[6] FreshLLMs: Refreshing Large Language Models with Search Engine Augmentation

**Questions For Authors:**

Please see the “Other Strengths And Weaknesses” section.

**Relation To Broader Scientific Literature:**

This paper relates to retrieval-augmented generation and reasoning abiltiies of long-context LLMs. It also relates to dataset and benchmark construction and evaluation.

**Theoretical Claims:**

Please see the “Other Strengths And Weaknesses” section below.

---

> ### Author Rebuttal · Authors · 2025-04-01
>
> We thank the reviewer for the thoughtful and encouraging assessment of PhantomWiki. We are pleased that you find PhantomWiki a high-quality benchmark for fine-grained evaluation of reasoning and retrieval. We are also delighted to hear that our work is **well-documented**, and presents **solid experimentation and analysis**. We address your concerns below and outline how we plan to revise the paper accordingly.
>
> **1. Multi-hop RAG baselines**
>
> > This paper could be improved by incorporating more recent baselines. e.g. IRCoT
>
> We add a **new experiment** with results of IRCoT from Trivedi et al. (2022) and Self-Ask from Press et al. (2022) in comparison to ZeroShot-RAG and CoT-RAG: see https://imgur.com/a/OMjtf3t. We would like to point out that **IRCoT and Self-Ask continue to struggle on questions with high difficulty**, measured by number of reasoning steps. We will include a more detailed comparison of multi-hop RAG to Agentic prompting techniques in our main paper.
>
> **2. Comparison to other existing benchmarks**
>
> > While using a context-free grammar and Prolog to generate question-answer pairs is intriguing, the concept of generating questions beyond the pre-training set of LLMs is not novel.
>
> We appreciate the reviewer’s observation regarding prior benchmarks that also generate dynamic or leakage-resistant question-answer pairs. However, we respectfully argue that **PhantomWiki is fundamentally different** in motivation, construction, and evaluative capabilities compared to these works:
> - ToolQA [4] focuses on tool-use evaluation, involving curated tools and questions that explicitly require external tools (e.g., calculators, databases). While it minimizes data overlap, its scope is about measuring LLMs’ ability to use predefined tools. In contrast, PhantomWiki evaluates **reasoning and retrieval in a self-contained world** and no external tools are assumed. The choice of external tools is left to the LLM and the prompting method.
> - REALTIME QA [5] and FreshQA [6] both target dynamic, real-world, up-to-date question answering, emphasizing temporal freshness and world knowledge. These benchmarks evaluate how well LLMs adapt to new information by querying real events or fast-changing facts. Our work takes the opposite approach: we **intentionally avoid real-world knowledge to enable clean disentanglement of reasoning and retrieval**.
>
> The core novelty of PhantomWiki lies **not just in generating unseen questions**, but in generating **entire fictional universes**, complete with articles, knowledge graphs, and questions. As you also point out, our datasets are **coherent**, and allow for **fine-grained control** over question difficulty and retrieval complexity—these functionalities distinguish PhantomWiki from [4,5,6]. LLMs can thus be evaluated on-demand on their ability to navigate and reason over an unseen yet structured universe.
>
> We appreciate the reviewer’s framing—you're right that our main contribution lies in the fine-grained controlled evaluation that fictional synthetic universes offer. We’ll make this positioning clearer in the revised manuscript and expand the related work section to reflect this comparison.
>
> **3. On Question Diversity and Domain Scope**
>
> > The generated questions tend to be less diverse, primarily focusing on reasoning between personal relationships and objects.
>
> We appreciate the reviewer’s point about the limited scope of the current question types, which primarily focus on relationships and attributes within a fictional universe. This design was intentional for the initial version of PhantomWiki, as it allowed us to **carefully control the reasoning complexity** and ensure data consistency and leakage resistance.
>
> That said, we completely agree that increasing the diversity of question types—especially to include temporal, causal, and dynamic aspects of the universe—would significantly expand the benchmark’s utility and better reflect real-world reasoning demands.
>
> Due to space constraints in this paper we are planning future work on extending PhantomWiki to support temporal components, such as characters aging, changing jobs, or forming new relationships over time. This will enable questions involving temporal reasoning, event sequences, and dynamic changes in the knowledge graph. These additions would naturally move PhantomWiki closer to the evolving and dynamic nature of benchmarks like REALTIME QA [5] and FreshQA [6], while still retaining the **key benefits of synthetic control and contamination resistance**.
>
> We’ll highlight these future directions more clearly in the discussion section of the revised manuscript, and we appreciate the reviewer for raising this valuable point.
>
> Thank you again for your detailed review, and please let us know if you have any more questions or concerns that could help improve the paper further!

---

> > ### Comment · Reviewer_GF7T · 2025-04-03
> >
> > Thank you to the authors for their efforts and thoughtful responses.
> >
> > I appreciate the inclusion of additional experiments on multi-hop RAG baselines. Indeed, when multiple retrievals are allowed, multi-hop RAG demonstrates slightly better performance than naïve RAG. This enhancement contributes to a more comprehensive evaluation of the proposed benchmark.
> >
> > I agree that RealTimeQA and FreshQA focus on dynamic world knowledge. While they also provide non-contaminated data for evaluation, their purpose differs from that of PhantomWiki. However, ToolQA evaluates not only LLMs’ ability to use tools but also their reasoning capabilities, as LLMs must determine which tools to utilize for different questions.
> >
> > While PhantomWiki generates entire fictional universes to enable fine-grained control over reasoning complexity, the current version's questions are primarily limited to relationships and attributes. Additionally, its approach to "controlling reasoning complexity" is based solely on recursion depth, which may not align with real-world scenarios. In reality, it is rare for questions to require reasoning over more than three or four chained relationships (in the paper, the maximum depth is 20). In contrast, the questions generated by ToolQA are more diverse and realistic, better reflecting real-world reasoning challenges.

---

> > > ### Author Response · Authors · 2025-04-08
> > >
> > > Thank you for your comments! We are happy to hear that the **new multi-hop RAG experiments have helped us address the first point in your initial review and strengthen our results**, and that **PhantomWiki differentiates itself well** from RealTimeQA and FreshQA through deliberate evaluation of **external knowledge-independent** reasoning abilities.
> > >
> > > We also agree with you that ToolQA has many strengths that complement those of PhantomWiki. Would like to expand on these two works and highlight the differences.
> > >
> > > **ToolQA**:
> > > * ToolQA has been initially designed to evaluate whether a model answers questions using tools or recalls a memorized answer. Its main strengths include a range of application domains, the use of (combinations of) tools, and diverse question templates (albeit in a relatively small number per application domain).
> > > * With one exception, ToolQA's underlying datasets are **static and memorizable**. At the time of publication (2023), the datasets were contemporary and therefore suitable for assessing tool use vs memorization. However, this selection was **short-term** and is now **at risk of becoming obsolete** for this purpose. For example, ToolQA's math questions were based on the "error cases made by ChatGPT" on GSM8K; it has since been demonstrated that commercial LLMs might be memorizing GSM8K's test case answers (cf. L44-49 of manuscript). We also believe that criteria like "information is too detailed for LLMs’ internal knowledge" (AirBnB; B.1 of ToolQA) will soon (if not already) become obsolete and in itself a **weak protection from data leakage**. Since ToolQA's datasets have been **manually curated**, any updates of ToolQA for next-generation LLMs is a nontrivial and yet short-term solution.
> > > * The only dataset underlying ToolQA that could in principle be non-static (Agenda) is generated using LLMs. As shown in our updated experiments, even rewriting a fixed set of facts can introduce inconsistencies which **need to be manually verified**; in their OpenReview submission, the authors of ToolQA mention reviewing the dataset for "3-4 rounds… to guarantee the validity", and that "it is quite **difficult to guarantee question quality**".
> > >
> > > **PhantomWiki**:
> > > * We instead focus on a more limited, but **more comprehensive and long-lasting methods for evaluation of retrieval and reasoning**.
> > > * We focus on making our benchmark **more resistant to updated LLM knowledge cutoffs** by introducing **on-demand, automatic** generation of new dataset versions **without any human involvement**.
> > > * We **intentionally stress-test** LLM reasoning and retrieval abilities through arbitrarily difficult reasoning questions. Just like we would expect a math LLM to be able to add **any** two numbers (not just the "real-world" two numbers, or a "real-world" number of sequential additions), we expect a true reasoning LLM to be able to do **multi-branch multi-step reasoning regardless of the number of steps**—as straightforward logic programs can answer these questions trivially. We would also like to emphasize that we control difficulty not just through the number of steps, but also by testing on **all possible solutions** to a question ("multi-branch reasoning"). We will make these points more clear in the revised manuscript.
> > > * We demonstrate that **frontier LLMs already struggle with PhantomWiki, despite its limited universe setting and despite its tasks being easily solvable** (for instance by logic programs). We appreciate and fully agree with your suggestions that PhantomWiki can be made even more diverse and realistic by, e.g., augmenting the universe with spatio-temporal data, introducing new types of entities and relationships (even arbitrary ones with *no* real-world meaning to completely disentangle internal knowledge from reasoning), and extending the context-free grammar to have an even larger diversity of verifiable QA templates. Due to space constraints and a substantial increase in methodological complexity of these additions—and PhantomWiki's value to the community even in its current version—we decided to postpone them to future work.

---

### Official Review · Reviewer_8LpG · 2025-03-14

**Overall Recommendation:** 4

**Summary:**

The manuscript introduces PhantomWiki, a generator for fictional universe of characters in the form of a fandom wiki. The knowledge graph for characters, their relations, and facts about them, are generated by sampling simple distributions. Articles are generated from these facts using templates. Questions are generated using templates, and answers are generated using Prolog solver on the original facts. Different size of fictional wiki can be generated using different numbers of characters, and easier/harder questions may be considered in terms of the number of "hops" required to answer them. PhantomWiki thus generates benchmarks involving novel "facts" to evaluate different language models and techniques in different regimes.

**Claims And Evidence:**

The manuscript does not provide an explicit list of claims, so I identified some myself.

### Claim 1: PhantomWiki can be tuned to benchmark different aspects/regimes

The benchmark can be made smaller or larger than a model's context length, and the questions can be made to require more or less documents to be answered. Varying these parameters yield consistent results on In-Context, RAG and Agentic strategies. Ignoring eventual Claim 3 issues (below), I judge that Claim 1 is supported by evidence.


### Claim 2: PhantomWiki is resistant to data contamination

By construction, models evaluated on PhantomWiki cannot memorize the exact answer to a question, nor any of the key information required in the intermediate steps, because this data is generated on-the-fly, immediately before evaluation.

However, once PhantomWiki is release in the wild, models can be trained, fine-tuned, etc. to perform better on PhantomWiki. Indeed, the probability distribution over the different generated datum can be learned, and learning how to best extract key information pieces from template-generated documents is easy for LLMs.

My assessment is that, while there exists a weaker claim that is supported, the claims made in the current manuscript are too broad. This is the topic of my Question 1 below.


### Claim 3: (Implicit) PhantomWiki is a good/useful benchmark that assesses aspects that other benchmarks don't assess

Assuming that there is no bug in the code, there should, by construction, be valid answers(s) planted in the dataset for each question (and the manuscript mentions that the case where there are multiple correct answers is handled properly). In a sense, the benchmark should be "more valid" than one that relies on human or LLM annotations.

However, although the manuscript presents experimental results for combinations of LLMs and reasoning/retrieval paradigms (In-Context, RAG, Agentic), PhantomWiki **itself** is not subjected to any direct evaluation of fitness for a purpose, nor are its scores related to existing benchmarks for consistency and/or redundancy assessment, nor is the score variability between two instances of PhantomWiki assessed.

My Question 2 below pertains to this topic.

**Essential References Not Discussed:**

Some long-context benchmarks:

- Hsieh et al. RULER: What’s the Real Context Size of Your Long-Context Language Models? COLM 2024. https://openreview.net/pdf?id=kIoBbc76Sy
- Wang et al. Leave No Document Behind: Benchmarking Long-Context LLMs with Extended Multi-Doc QA. EMNLP 2024. https://aclanthology.org/2024.emnlp-main.322.pdf
- Zhang et al. ∞Bench: Extending Long Context Evaluation Beyond 100K Tokens. ACL 2024. https://aclanthology.org/2024.acl-long.814.pdf

On creating the corpus:

- Monteiro et al. RepLiQA: A Question-Answering Dataset for Benchmarking LLMs on Unseen Reference Content. NeurIPS 2024. https://proceedings.neurips.cc/paper_files/paper/2024/file/2b23626015b6311369e95a70735cbb72-Paper-Datasets_and_Benchmarks_Track.pdf

Knowledge graphs:

- Agarwal et al. Knowledge Graph Based Synthetic Corpus Generation for Knowledge-Enhanced Language Model Pre-training. ACL:HLT 2021. https://aclanthology.org/2021.naacl-main.278.pdf
- Ye et al. Generative Knowledge Graph Construction: A Review. EMNLP 2022. https://aclanthology.org/2022.emnlp-main.1.pdf

**Experimental Designs Or Analyses:**

The experimental design uses the introduced benchmark to assess different LLMs in different retrieval/reasoning modes. For that purpose, the experimental design appears decent to me. However, as mentioned before, there are no direct experiments evaluating the benchmark itself.

**Methods And Evaluation Criteria:**

The results presented make sense on their own. However, the purpose of the manuscript is to introduce a benchmarking method, and that benchmarking method itself is not directly evaluated. See my comments on Claim 3 and Question 2.

**Other Comments Or Suggestions:**

> friendship graph using the Erdős–Rényi model

Friendship (social) networks, real and fictional, diverge in many way from this model. In my personal opinion, the two main aspects worth capturing are transitivity ("I am more likely to be friend with the friend of my friend than with with a random person") and heavy-tailed degree distribution ("most people have few friends, few people have lots of friends"). If the authors are interested to learn more, they may start with https://en.wikipedia.org/wiki/Triadic_closure and https://en.wikipedia.org/wiki/Scale-free_network for these two aspects, respectively, and https://en.wikipedia.org/wiki/Social_network_analysis for a more general discussion. No actual changes are requested on this manner: these issues are very weakly coupled with the manuscript's concrete goals.

**Other Strengths And Weaknesses:**

The core idea of automatically generating a benchmarking corpus, with associated questions and answers, parametrized in terms of corpus size and question complexity, is a promising one.

**Questions For Authors:**

### Question 1: Do you have a solution to propose for my concerns with Claim 2?

One possible avenue is to tone down the resistance to data contamination claims. Another avenue is to try to actually fine-tune models on multiple instances of PhantomWiki, and assess how this affects performances on fresh instances. In any case, please state what edits would be made to the manuscript.

### Question 2: Can you provide more direct evidences that PhantomWiki provides actionable assessments of language models?

One possible avenue is to relate the results on PhantomWiki to pre-existing benchmarks and/or human evaluations. Other avenues involve designing and running new experiments, but I understand that time is short. Finally, perhaps I've missed something obvious, and all you have to do is to explain it better to me.

In any case, please keep in mind the following question "**What would negative results look like?**". In the (potentially counterfactual) scenario where PhantomWiki turned out to not be a useful benchmark, what different observations would have been made? I acknowledge that the narrative consistency of results mentioned in Claim 1 is weak evidence in PhantomWiki's favour, but I'm looking for something more direct.

### Question 3: Could you provide examples of a small corpus?

Say, generate $n=4$ articles in an appendix, with some associated question/answer pairs? Figure 2 (2) is very minimal...

**Relation To Broader Scientific Literature:**

The related work section focuses on agent and/or tool-use evaluation benchmarks, but neglects "long-context" ones. Some suggestions are provided in the next section.

> Importantly, none of these benchmarks creates the underlying corpus, a limitation which we bridge in this work.

∞Bench (Zhang et al., 2024) uses a key entity replacement strategy to this end. However, it is subjected to the "Beethoven might have met Mozart" failure mode mentioned by the authors. RepLiQA (Monteiro et al. 2024) have humans generate the corpus. However, the questions can all be answered using a single document.

This work could be framed as the generation and verbalization of a knowledge graph. Some relevant works are listed below.

**Theoretical Claims:**

The description of how the original graph/data is generated makes sense, though is quite simplistic. The use of a single template to generate all entries is very disputable (and part of my concerns in Claim 3), especially considering that this template basically amounts to concatenate sentences of the form "The X of Y is Z.". The generation of questions and answers make sense, though I did not verify in depth the grammars and templates.

---

> ### Author Rebuttal · Authors · 2025-04-01
>
> **1. Question 1 / Evaluating language models fine-tuned on PhantomWiki data**
>
> > Another avenue is to try to actually fine-tune models on multiple instances of PhantomWiki, and assess how this affects performances on fresh instances.
>
> We add a **new experiment** and find that **fine-tuning on PhantomWiki data helps** performance on held-out instances, but only **to a limited extent**: see https://imgur.com/a/QHftpCM (please see also our rebuttal to reviewer YaUR for full experiment details). Both prompting and fine-tuning approaches struggle as question difficulty increases. Thank you for suggesting this experiment idea! We are excited to see how PhantomWiki inspires research in LLM reasoning.
>
> **2. Response to Claim 3 and Question 2**
>
> > PhantomWiki itself is not subjected to any direct evaluation of fitness for a purpose, nor are its scores related to existing benchmarks for consistency and/or redundancy assessment, nor is the score variability between two instances of PhantomWiki assessed.
>
> We appreciate your framing of “what would negative results look like?”, which we find especially helpful for clarifying our benchmark’s utility. Below, we outline how PhantomWiki provides actionable assessments and how we plan to revise the manuscript.
>
> A poorly designed benchmark might fail to appropriately differentiate models and/or distinguish between reasoning paradigms. Our experiments suggest the opposite: **PhantomWiki differentiates performance** based on both model architecture and prompting techniques, and **elucidates failure modes** (e.g., hallucinated intermediate hops or context retrieval mismatches) that align with known model weaknesses.
>
> Based on our evaluation results, PhantomWiki is _not_ a good benchmark for reading comprehension, as evidenced by the near perfect F1 scores of all LLMs/prompting techniques for questions that require 1 reasoning step. Indeed, PhantomWiki is meant to complement more complex reading comprehension benchmarks like DROP and SQuAD. Nonetheless, it is a good benchmark for evaluating **multi-step and multi-branch reasoning**. Quantitatively, we can see this from the rapid drop in performance in Figure 3 as the number of reasoning steps increases.
>
> One area where LLMs struggle is with complex relations. For example, Llama-3.3-70B-Instruct struggles with “Who is the great-grandchild of X?”, not because it doesn’t know what great-grandchild means, but rather because of a high step count (child of child of child) and a high branching factor (having to find the children of each grandchild). One actionable insight is to improve language models’ ability to keep track of not only multiple steps (this is corroborated by the Self-Ask paper, which highlights a compositionality gap), but also multiple branches. PhantomWiki thus **sheds light on both of these challenges** in LLM reasoning. We will include the discussion of failure modes of different models in the revision.
>
> In terms of benchmark stability, we run all the evaluations with the same hyperparameters on the first three seeds of generated PhantomWiki instances and **measure variability** through standard deviation across runs. This provides evidence that **PhantomWiki is robust** to sampling variance and not overly sensitive to individual instance characteristics.
>
> > The use of a single template to generate all entries is very disputable (and part of my concerns in Claim 3), especially considering that this template basically amounts to concatenating sentences of the form "The X of Y is Z."
>
> **In a new experiment, we use Llama-3.3-70B-Instruct to rephrase our templated articles** (see prompts: https://imgur.com/a/ffvugvg and example generations: https://imgur.com/a/wFvjAFP), which we then use for downstream evaluation: see https://imgur.com/a/DQuvQIA (please see also rebuttal w1Q5 for further details). We report similar performance trends as in Figure 3 from our manuscript, and in fact, using rephrased articles only makes PhantomWiki more challenging. Thus templated articles have the benefit of being cheaper, faster, and free of LLM hallucinations, while still providing valuable insight into LLM reasoning capabilities. Importantly, PhantomWiki is designed to be modular to incorporate LLM-generated articles, as future research in reducing LLM hallucination matures.
>
> **3. Additional references on long-context and friendship graphs**
>
> We greatly appreciate the reviewer’s suggestion of connecting to the long-context literature. We will revise the Related Works section to include them. We also plan to support more realistic friendship graphs, especially heavy-tailed degree distributions, in future versions of PhantomWiki.
>
> **4. Example of a small universe**
>
> Please see https://imgur.com/a/QLhC54R.
>
> We hope the new experiments and discussion addressed your concerns and strengthened our results, and if so we would like to politely ask you to consider raising your score. Thank you again!

---

> > ### Comment · Reviewer_8LpG · 2025-04-03
> >
> > Thank you for the clarifications.
> >
> > With the expectation that the camera ready will be adapted accordingly, including mentions in the introduction and/or abstract as to what the benchmark is good at evaluating (e.g., evaluate multi-step multi-branch reasoning) and what it isn't (e.g., reading comprehension), I hereby increase my Overall Recommendation from 2 to 4.

---

### Official Review · Reviewer_YauR · 2025-03-14

**Overall Recommendation:** 3

**Summary:**

The paper presents PhantomWiki, a pipeline that generates synthetic large-scale document corpora and question-answer pairs. PhantomWiki works by generating articles containing facts about characters (e.g., “The job of x is y.”), and generating question-answer pairs from templates. PhantomWiki allows generating large corpora, exceeding the context length of current LMs. The paper experiments with in-context prompting, RAG, and agentic RAG models on PhantomWiki, and demonstrates that current models struggle with large corpora and complex questions generated using the PhantomWiki pipeline.

**Claims And Evidence:**

While the paper has merits, and I believe the proposed method can be a promising alternative to the popular “needle-in-a-haystack” experiments, I believe that the claim that the current method is reliable in evaluation of RAG models is problematic.

My main concern regards the templated data generation approach. Specifically, the distribution of the generated documents (e.g., “The job of x is y.”) and questions (templates are presented in B.2), seems to be very different from realistic RAG settings. Second, as the data was generated via templates, I am concerned it will be easy to game with additional training (for example by using the same templates to generate training data).

**Essential References Not Discussed:**

I believe that the paper can benefit from a discussion regarding multi-hop RAG (e.g., IR-CoT), and synthetic data generation from KG (many methods use a KG, e.g., Wikidata, and then generate questions synthetically using the relations).

**Experimental Designs Or Analyses:**

I checked the soundness of the experiments in Sections 4-6.

**Methods And Evaluation Criteria:**

I believe the paper could benefit from additional multi-hop RAG basslines (e.g., IR-CoT, Self-Ask).

Additionally, as the paper proposed a new evaluation method for RAG, it will be helpful to evaluate the gains from fine-tuning on the synthetic distribution (see Claims and Evidence).

**Other Comments Or Suggestions:**

None

**Other Strengths And Weaknesses:**

The paper is overall well-written and easy to follow.

**Questions For Authors:**

None

**Relation To Broader Scientific Literature:**

By generating a large corpora with matching question-answer pairs that is robust to data contamination teh paper proposes an interesting evaluation method for long-context reasoning, which has several benefits over previous approaches (e.g., “needle-in-a-haystack”). The paper also compares in-context prompting, RAG, and agentic-RAG models, showing all approaches struggle as the number of reasoning steps increases.

**Theoretical Claims:**

None.

---

> ### Author Rebuttal · Authors · 2025-04-01
>
> Thank you for your thoughtful comments and suggestions, which we believe improve our manuscript significantly. We are pleased that you find PhantomWiki a **promising alternative to the popular “needle-in-a-haystack” test**. We address your comments below and will add results for suggested experiments to our manuscript.
>
> **1. Use of templated articles**
>
> Thank you for pointing this out. **We have added an experiment where we use Llama-3.3-70B to rephrase our templated articles** (see prompts: https://imgur.com/a/ffvugvg and example generations: https://imgur.com/a/wFvjAFP) and evaluated on these rephrased PhantomWiki corpora: see https://imgur.com/a/DQuvQIA. We find the **drop in F1 scores versus question difficulty to be similar** to our original experiments (Table 2 of the manuscript). On the other hand, templated articles are cheap, fast, and free of LLM hallucinations, while enabling compelling evaluations for LLM reasoning capabilities. Importantly, PhantomWiki is designed to be modular to incorporate LLM-generated articles, as future research in reducing LLM hallucination advances.
>
> **2. Evaluating LLMs finetuned on PhantomWiki data**
>
> > Second, as the data was generated via templates, I am concerned it will be easy to game with additional training (for example by using the same templates to generate training data).
>
> > Additionally, as the paper proposed a new evaluation method for RAG, it will be helpful to evaluate the gains from fine-tuning on the synthetic distribution (see Claims and Evidence).
>
> Thank you for suggesting this experiment. As the relationships, names, and attributes in each PhantomWiki universe are generated randomly on-demand, we believe that naive training on PhantomWiki data can only yield limited improvements. To support this claim, we have a **new experiment** where **we generate 10 new PhantomWiki dataset instances** (question depth 20 and universe size 50) amounting to 5K training question-answer pairs. We then **perform full fine-tuning** of Qwen2.5-0.5B-Instruct and parameter-efficient fine-tuning of Qwen2.5-3B-Instruct with LoRA on all linear layers.
>
> For each base model, we employ two popular training algorithms. The first is Group Relative Policy Optimization (GRPO) from [Shao et al.](https://arxiv.org/abs/2402.03300) with an F1-score reward between 0 and 1. We use the CoT prompt template from App. C.4, a batch size of 32, and 8 generations per prompt to sample. The second is supervised fine-tuning (SFT) with answer(s) as the ground-truth label. We use the zeroshot prompt in App. C.2 and a batch size of 4. For all training experiments, we train for 3 epochs (or until convergence) using the AdamW optimizer with initial learning rate set to $5\times 10^{-6}$ for full fine-tuning and $10^{-4}$ for LoRA fine-tuning.
>
> We then evaluated the fine-tuned models using the PhantomWiki instances of size $n=50$ from Table 2: see https://imgur.com/a/QHftpCM. For Qwen2.5-0.5B, we find that GRPO and SFT both improve F1 compared to prompting-based methods, likely due to improved ability to output the proper answer format. For Qwen2.5-3B, we find that GRPO improves F1 slightly, whereas SFT worsens F1, likely due to overfitting on the training samples. These experiments show that **further advances beyond fine-tuning are needed to truly close the gap on PhantomWiki**; we hope that PhantomWiki will serve as a valuable tool for future research on LLM reasoning and retrieval.
>
> **3. Additional multi-hop RAG baselines e.g. IRCoT and Self-Ask, discussion on synthetic data from KG**
>
> Following your suggestion, **we have performed additional evaluation using IR-CoT and Self-Ask** with BM25 as the retriever and Llama-3.3-70B-Instruct as the generator model: see https://imgur.com/a/OMjtf3t. Specifically, we used the implementations from FlashRAG ([Jin et al. 2024](https://github.com/RUC-NLPIR/FlashRAG)) and re-wrote the few-shot examples where needed to match the formatting of PhantomWiki questions. We find that both IR-CoT and Self-Ask are competitive with ZeroShot-RAG and CoT-RAG. However, when decomposing model performance on question difficulty (like Figure 3), **both IR-CoT and Self-Ask struggle like other prompting methods**. In fact, this is a key contribution of PhantomWiki: quantitatively decomposing performance of prompting methods and LLMs on axes of reasoning and retrieval.
>
> Notably, IR-CoT and Self-Ask alternate between reasoning and retrieval like agentic prompting (e.g., ReAct). A key difference: ReAct uses LLM-driven retrieval via tools, while multi-hop RAG methods delegate retrieval to an external retriever—decoupling reasoning and retrieval. We’ll highlight this nuance in the revised manuscript. We’ll also add a brief discussion about generating data from knowledge graphs.
>
> Given these **new experiments** support the paper's message and **strengthen our empirical results**, we would like to politely ask you to consider increasing your score. Thank you again!

---

### Decision · Program_Chairs · 2025-05-01

**Decision:**

Accept (poster)

**Comment:**

The paper presents a synthetic benchmark for evaluating multi-hop RAG. Reviewers appreciated the scale of the dataset and the control over the difficulty of examples generated. During the rebuttal, addition of new baselines and a paraphrased version was also useful to understand the utility of the benchmark. However, the fact remains that the dataset constructed may not be realistic and contain template-like relationships. As Rev. 8LpG suggests, it will be good to highlight in the intro the kind of tasks for which the benchmark is well-suited for.
On balance, the paper presents a useful addition to evaluating RAG systems.